# How water, temperature and seismicity control the preconditioning of massive rock slope failure (Hochvogel)

Johannes Leinauer[1], Michael Dietze[2,3], Sibylle Knapp[1,4], Riccardo Scandroglio[1], Maximilian Jokel[1], and Michael Krautblatter[1]

[1]Technical University of Munich, TUM School of Engineering and Design, Landslide Research Group, München, Germany
[2]Georg-August-Universität Göttingen, Faculty of Geosciences and Geography, Göttingen, Germany
[3]GFZ German Research Centre for Geosciences, Potsdam, Germany
[4]UNESCO Global Geopark Swabian Alb, Schelklingen, Germany

**Correspondence:** Johannes Leinauer (johannes.leinauer@tum.de)

**Abstract.** The anticipation of massive rock slope failures is a key mitigation strategy in a changing climate and environment requiring a precise understanding of pre-failure process dynamics. Here we exploit >4 years multi-method high-resolution monitoring data from a large rock slope instability close to failure. To quantify and understand the effect of possible drivers - water from rain and snowmelt, internal rock fracturing and earthquakes - we correlate slope displacements with environmental data, local seismic recordings and earthquake catalogues. During the snowmelt phase, displacements are controlled by meltwater infiltration with high correlation and a time lag of 4-9 days. During the snow-free summer, rainfall induces accelerations with a time lag of 1-16 h for up to several days without a minimum activation rain sum threshold. Rock fracturing, linked to temperature and freeze-thaw cycles, is predominantly surface-near and unrelated to displacement rates. A classic Newmark analysis of recent and historic earthquakes indicates a low potential for immediate triggering of a major failure at the case site, unless it is already very close to failure. Seismic topographic amplification of the peak ground velocity at the summit ranges from a factor of 2-11 and is spatially heterogeneous, indicating a high criticality of the slope. The presented in-depth monitoring data analysis enables a comprehensive rockfall driver evaluation and indicates where future climatic changes, e.g. in precipitation intensity and frequency, may alter the preconditioning of major rock slope failures.

## 1 Introduction

Massive rock slope failures are an important geomorphic hazard (e.g. Evans et al., 2006; Lacasse and Nadim, 2009) and in the wake of climate change, landslide risk is expected to increase in many regions (e.g. Gariano and Guzzetti, 2016; Picarelli et al., 2021). To prevent damage to people or property, the anticipation of such events becomes highly crucial (Sättele et al., 2016; Chae et al., 2017; Pecoraro et al., 2019; Leinauer et al., 2023), and thus, relevant drivers and potential triggers of imminent failures must be identified and understood. Currently, the capacity to monitor all hazardous rock slopes in a way that allows site-specific process analyses is not existent. We must therefore rely on standard and qualitative rock fall triggering factors, or better, quantify the relevant drivers based on multi-method high-resolution monitoring data at well-equipped sites. Exploiting the available data of comprehensive monitoring and early warning systems to gain understanding of all relevant

pre-failure process dynamics should therefore become a standard procedure (Gischig et al., 2016), allowing the inference of trigger anticipation strategies for similar slopes.

Rockfall release can be caused by a reduction of resisting forces and/or an increase of driving forces. In the preparation phase, promoting drivers act on a rock slope over months to millions of years (Dietze et al., 2017b) bringing the system progressively closer towards critical slope stability (Oswald et al., 2021). This is achieved by the development of a sliding plane over different time scales, e.g. through seasonal pore pressure increase (Preisig et al., 2016), multi-year seismic loading (Gischig et al., 2016), or long-term fracture propagation following weathering and erosion, glacial debuttressing (Eberhardt et al., 2004; Ballantyne

et al., 2014) or permafrost degradation (Hilger et al., 2021). Finally at failure, a trigger acts on the balance between stabilising forces and stress, leading to unstable conditions initiating rockfall within short time (Wieczorek, 1996). Of course, promoting drivers in the preparation phase and triggering factors terminating this phase can overlap and interact, and the transition between the two may be gradual. In some cases the progressive weakening of material could lead to slope failure without an apparent external trigger (Lagarde et al., 2023) but in such state, the rock slope instability becomes increasingly sensitive to external

drivers. However, a detailed and comprehensive knowledge of how and how much internal and external drivers control the pre-failure stage of imminent rock slope failures is missing at most sites, but crucial for anticipation tasks.

  Possible rockfall drivers and triggers (Fig. 1) include (e.g. Stock et al., 2013; Dietze et al., 2017b) (a) rainfall, (b) snowmelt, (c) rock fracturing and crack propagation, (d) earthquakes, (e) temperature gradients, (f) freeze-thaw-cycles, (g) wind, (h) light-ning, (i) rock or ice fall inducing secondary rockfall, (j) snow or rock avalanches, (k) volcanic activity, (l) vegetation growth

and root prying, (m) permafrost degradation (Krautblatter et al., 2013), and (n) human or animal activity. The significance of some of these factors might change in the future due to climatic fluctuations. Temperature and precipitation patterns are expected to change in many regions (IPCC, 2019), expressed by a general warming, modified extreme rain events, or snow cover shifts (e.g. Huss et al., 2017; Pendergrass et al., 2019). This influences the timing and amount of available water in the system or rock/ ice properties and thus, the sensitivity of affected drivers to climate change should be evaluated (Agliardi et al.

(e.g. 2020), cp. Section 4.6).

  Heavy precipitation and rapid snowmelt are documented to be amongst the most important drivers for rockslides across the globe (e.g. Wieczorek, 1996; Helmstetter and Garambois, 2010; Stock et al., 2013; LaHusen et al., 2020). Infiltrating water can destabilize rock slopes repeatedly in short time scales by building hydrostatic pressure in fractures, elimination of the joint cohesion, lowering of the joint friction angle and reduction of the effective normal stress at the sliding surface due to

uplift (Erismann and Abele, 2001; Wyllie and Mah, 2004; Scandroglio et al., 2021). Especially in spring, large amounts of water can infiltrate into slopes due to simultaneous intense snowmelt and rainfall (Kawagoe et al., 2009; Krøgli et al., 2018; Lorenzi et al., 2024). Additionally, long-term hydro-mechanical loading cycles are proven to have a promoting effect in deep-seated landslides (Gischig et al., 2016). Constraining the amount of water, specifically the pore pressure inside a rock slope, is challenging without the availability of boreholes. In such cases, the direct measurement of the water supply via rain gauges or

snow stations is easier to obtain.

  Progressive crack propagation in fracturing rock is one of the main internal drivers of rock slope failures (Petley, 2004; Lagarde et al., 2023). Critical and subcritical crack growth along active sliding planes intensifies the stress concentration at

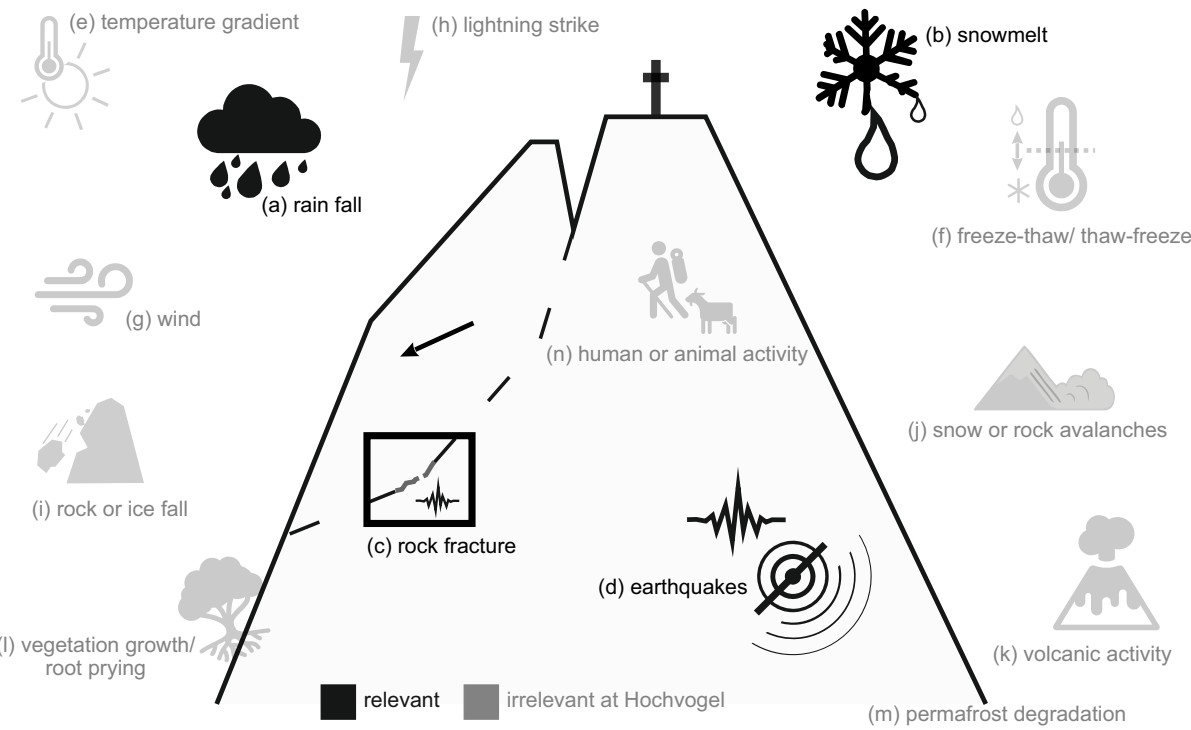

**Figure 1.** Factors that can potentially promote and/or trigger rockfalls. Black processes (a-d) are relevant for a major failure at our case site Hochvogel, grey ones (e-n) can be excluded due to the reasons in the text.

the crack tips (e.g. Amitrano and Helmstetter, 2006; Voigtländer et al., 2018), as with every failing rock bridge, the stress increases at the remaining rock bridges (Kemeny, 2003). Once most rock bridges have been degraded, episodic deformation
release might be controlled by macro-roughness creating obstacles along the sliding plane (Borri-Brunetto et al., 2004; Dietze et al., 2021). Stress release in the form of rock fracturing generates typical and distinguishable seismic signals that can be recorded with local seismic networks (Senfaute et al., 2009; Helmstetter and Garambois, 2010; Hibert et al., 2011; Provost et al., 2017; Dietze et al., 2021; Lagarde et al., 2023). Isolating the episodically occurring short events within the large datasets is challenging with manual techniques and thus, established machine learning procedures can help to build a database of rock
fracturing events (Provost et al., 2017; Hibert et al., 2017; Wenner et al., 2021; Langet and Silverberg, 2023).

    Earthquakes frequently trigger numerous landslides (e.g. Wieczorek, 1996; Jibson et al., 2006; Meunier et al., 2007; LaHusen et al., 2020; Marc et al., 2016; Massey et al., 2022). They seem to have played a significant role in preparing and triggering prehistoric large rockslides due to the spatio-temporal coincidence of major earthquakes and rock slope failures (Knapp et al., 2018; Oswald et al., 2021). The correlation of landslide occurrence and volume with peak ground acceleration (PGA) val-
ues (Meunier et al., 2007; Massey et al., 2022) reinforces the destabilizing nature of strong seismic waves, but recurring earthquakes are further a promoting factor due to seismic fatigue (Gischig et al., 2016; Oswald et al., 2021). Earthquake waves add

additional stress to slope instabilities by accelerating the ground with a specific amount of energy for a limited time depending on frequency and direction. This ground motion can be measured directly at an instrumented instability with the same local seismometers that also record rock fracturing. If direct observations at the site of interest are not available, theoretical block

displacements can be estimated with the well-known Newmark analysis (Newmark, 1965). This method allows to evaluate historic earthquakes from earthquake catalogues or systematic parameter sets towards their triggering potential and deformational influence. In steep topography, it is furthermore important to consider frequency-dependent seismic wave amplification due to topographic site effects (Harp and Jibson, 2002; Sepúlveda et al., 2005; Lee et al., 2009a, b; Khan et al., 2020). Through topographic resonance and refraction of waves, seismic amplification can reach factors of 2-14 in the horizontal component at

specific frequencies, predominantly triggering landslides at mountain tops and ridge crests facing away from the epicenter (Meunier et al., 2008; Bakun-Mazor et al., 2013; Rault et al., 2020; Weber et al., 2022). Moreover, seismic waves can be amplified and polarized within unstable rock mass itself due to existent open cracks, mainly perpendicular to them (Burjánek et al., 2010, 2012). Gischig et al. (2016) found modelling evidence that amplification factors increase and become more complex in space and frequency with a higher degree of slope damage, which may in turn be used for assessing the slope´s criticality.

However, comprehensive site-specific analyses of how seismicity controls the preparation phase of rock slope failures based on field observations and historical earthquakes are usually not performed.

In this study, we focus on massive rock slope failures, i.e. >20,000 m$^3$ (Evans et al., 2006). We use a well-prepared high-magnitude alpine rock slope instability in dolomite rock at the summit of the Hochvogel mountain (2592 m a.s.l., see details in Section 2), where we conducted multi-method high-resolution monitoring for more than four years. Due to the magnitude,

location and altitude of the rock slope instability, some generic drivers or triggers can be supposed irrelevant (Fig. 1). Strong temperature gradients and freeze-thaw cycles can only affect surface-near rock mass (Bakun-Mazor et al., 2013; Weber et al., 2017) without reaching deeper-seated sliding zones. The effect of wind is mostly connected to trees with roots (Stock et al., 2013; Dietze et al., 2015) which are absent at the Hochvogel, seismic noise (Lott et al., 2017), and pressure differences at the rock surface (Stock et al., 2013), which do not reach deep-seated sliding zones. Thus, wind can potentially induce small-

scale rockfall, but significant influence on several thousand cubic meters of rock is unrealistic. The same applies to lightning strikes. Excluding large-scale slope engineering, this likewise holds for human or animal activity. As the Hochvogel rock slope instability is located at the summit of the mountain, no rock or ice fall can impact on the instability. The same applies to snow or rock avalanches. Volcanic activity is absent in this region. Due to its altitude, the Hochvogel summit is above the treeline but below the permafrost limit. Thus, four relevant drivers remain and are therefore extensively analysed in this study: (i) rainfall-

provided water, (ii) snowmelt-derived water, (iii) stress-induced rock fracturing and (iv) seismic acceleration. Here, we analyse displacements as the phenomenological result of all drivers and correlate them with (i+ii) meteorological data, (iii+iv) local seismic recordings and (iv) earthquake catalogues. This approach allows to quantitatively evaluate promoting and triggering factors for many massive rock failures with a set of triggering conditions similar to our case study.

## 2   Study site and instrumentation

The Hochvogel (2592 m a.s.l.) is an isolated mountain peak with high topographic prominence in the eastern Allgäu Alps on the border between Germany and Austria (Fig. 2a). It consists of brittle, well layered and deformed dolomite rock (Hauptdolomit) from the Upper Triasic Lechtal nappe with incidental marly interlayers. The small-scale variation of rock properties is mainly primary due to the layering, but minor faults and folds add additional spatial heterogeneity. The summit area is characterized by a 2-6 m wide main crack that divides the massif into a stable NE-side and an unstable SW-side. The total unstable volume sums

up to 260,000 m$^3$ above a distinct 1 m thick marly layer (Fig. 2b-A, Leinauer et al. (2020)). Including a potentially unstable mass below, the total volume reaches 400,000-600,000 m$^3$. The rock slope instability has been developing at least since the 1940s with higher deformation rates of about $2\,\mathrm{cm\,a^{-1}}$ during the last two decades and is currently preparing to (partially) fail. Several lateral cracks at the almost vertical SW-wall have shown higher activity in the last decade. This flank shows frequent failure of rock towers, as for example a 130,000 m$^3$ rockfall in 2016 (Fig. 2b-B, Barbosa et al. (2024)). The site has been under

comprehensive monitoring since 2018/19 (Leinauer et al., 2021; Dietze et al., 2021), including observation of crack opening, temperature, rain and seismic signals. A detailed description can be found in Leinauer et al. (2020).

In this study, we exploit several high-resolution data from between October 2018 and November 2022. This includes displacements measured as crack opening by a vibrating wire crackmeter (Crack06) at the most active lateral crack (see photos in Fig. S1 and S2). Measurements are available with 10 min frequency from our real-time monitoring system (resolution

0.04 mm, accuracy $\pm$ 0.15 mm). With the same frequency, the air temperature is measured directly at the crackmeter and rainfall is measured with a non-heated tipping bucket rain gauge (resolution 0.1 mm, see photo in Fig. S3).

To monitor seismic processes, we installed a local seismic network on the summit consisting of 4 PE6B 4.5 Hz 3-component geophones and Digos DataCube3ext loggers (maximum distance 75 m). Three of the sensors ($SA_{21}$, $SA_{22}$, $SA_{23}$) are on the stable side, and one station ($HV_1$) is on the unstable side next to the main crack (Fig. 2b, see photos of the seismic stations in

Fig. S4 and S5). Additionally, we have installed a wider network with spacings between 0.9 and 1.6 km on the SW flank below the summit, consisting of TC120s or PE6B geophones and DataCube3ext or DataCube3extBOB loggers (Fig. 2a). All stations recorded ground velocity values at 100-400 Hz, but signals have been uniformly aggregated to 100 Hz before further analysis. The station meta data are listed in Table S1.

Data for snowmelt modelling come from two stations of the Bavarian Avalanche Warning Service in the region of the

Hochvogel (yellow diamonds in Fig. S26) that measure all necessary parameters (wind, surface and air temperature, snow height). The Nebelhorn (2075-2220 m a.s.l.) station is 9 km away from the Hochvogel, the Zugspitze station (2420-2960 m a.s.l.) 41 km.

To analyze the effect of local or regional earthquakes, we exploit the earthquake event catalogues of Germany (BGR, 2023) and Switzerland (SED, 2023) containing all registered regional earthquakes with $M_w$ >2 since 1692 and 250 AD, respectively.

Continuous observations with sub-second resolution are available since 1975 and 2009, respectively. To study the effect of strong distant earthquakes during station operation of our local seismic network, we use the catalogue of the US Geological Survey (USGS, 2023).

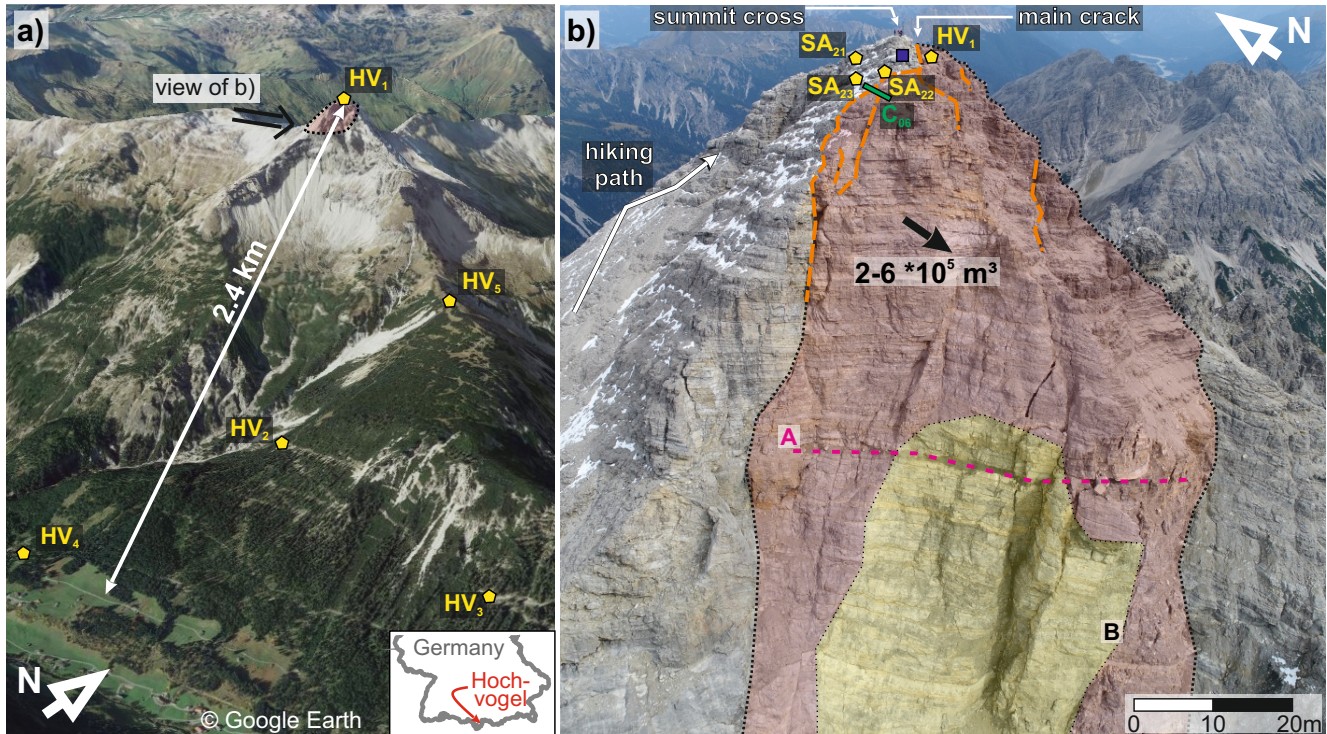

**Figure 2.** Overview of the study site. (a) Hochvogel mountain with the unstable mass at the summit (red area) and the seismic stations of the flank network (yellow pentagons). The village Hinterhornbach (AT) in the valley below the Hochvogel is in the bottom left of the image. Image source: © Google Earth. (b) Photo of the summit area and the steep SW-wall. The unstable mass is marked with the red area, obvious cracks are traced by dashed orange lines. Measurement devices in this study include the stations of the seismic summit network (yellow pentagons), rain gauge (blue square) and crackmeter "C06" (green line). A distinct 1 m thick marly layer is marked by the dashed magenta line (A). The source area of a 130,000 m³ rockfall from 2016 is marked by the yellow colour (perimeter B).

## 3 Data processing

All data were processed with the software R v. 4.3.0 (R Core Team, 2023). R scripts of all major processing and analysis steps are available in an online repository under https://doi.org/10.5281/zenodo.10567098 (Leinauer, 2024). For the availability of the underlying data see the code and data availability statement.

### 3.1 Seismic crack events

All seismic analysis was performed with the R package eseis v. 0.7.3 (Dietze, 2018a, b). To get corrected ground velocity time series (m/s), the seismic data were deconvolved according to each instrument specifications (see Table S1). Using the data from the summit network, we isolated discrete seismic events. We selected all times during which at least two stations operated simultaneously (860 days, 60 % coverage) to include only events that were captured by more than one station. The basic picking

routine then followed the approach by Dietze et al. (2021), using a classic short-term average/ long-term average (STA-LTA) ratio picker (Allen, 1882), applied to the 20–40 Hz filtered signal envelopes (on-ratio = 6, off-ratio = 1, STA-window = 0.2 s, LTA-window = 120 s). With these settings, even low-energy events were detected, but also many false positives. The following
automatic check of potential events premises an event duration of 0.2-5 s as the typical duration of discrete rock fracturing activity at the summit (e.g. Senfaute et al., 2009; Dietze et al., 2017a, 2021). Furthermore, the detection time difference between two stations needs to be less than 0.3 s according to a conservative wave travel time across the entire network. This excludes signals with longer detection time differences that result from the coincidence of unrelated signals or waves initiated by sound travelling through air. This led to a detection of 109,492 picked potential events for which we plotted seismograms
and spectrograms. A meaningful localization of of the signal source was hindered due to the steep and complex topography, highly jointed rock mass and unclear wave velocity distribution (cp. Helmstetter and Garambois, 2010).

Finally, to sort the picked events into two groups of seismic rock fracturing events (= target) and other events (steps of humans, rockfalls, coincident noise, false detections), we filtered the events further and used machine learning with a Random Forest classifier (Breiman, 2001), that has been developed in multiple processing steps. First, we calculated the signal-to-noise
(SNR) ratio and re-defined all start and end times of the events with a kurtosis picker (Baillard et al., 2014; Hibert et al., 2017), that can identify the picked onset more accurate than the robust STA/LTA picker used in the first step. We then rejected events without distinct ending where the picker could not identify a sharp decrease of the signal (4,813 events, usually humans walking next to the stations or signals with >5 s duration or high background noise not connected to cracking). We further rejected events where the kurtosis picker did not trigger (1,128 events), where the two picking routines identified start times
differing more that 1 s (2,071 events) and events where the kurtosis-picked event ended before the start of the STA/LTA-pick (12,560 events, usually when there was not one discrete isolated event). The Random Forest Classifier requires a set of features that describe the seismic signal and allow to separate between the different classes. We followed and adapted the approach by Hibert et al. (2017) and calculated a set of 61 statistical values from the waveform, spectral and pseudo-spectrogram domains. We did this for the station with the highest SNR at each of the remaining 88,920 events, once for the event itself and once
for a longer signal including 3 s buffer before and after the picked signal start and end times. Together with maximum SNR, minumum SNR, mean duration and the duration difference between the stations per event, we used a total number 124 features as Random Forest input (see details in the Table S2).

In a first step, we created a training data set by manually classifying 1,353 events (205 crack events, 1,148 others) looking at the features in the plots described in Dietze et al. (2021). We used a balanced proportion of 80 % of the data for
training and validating implementing a 5-fold cross-validation. We set up a Random Forest with 500 decision trees including hyperparametertuning with random search and 1000 iterations on the minimum size of terminal nodes, the maximum number of terminal nodes and the number of variables randomly sampled as candidates at each split (see the code under https://doi.org/10.5281/zenodo.10567098 for details). In the last step we tested the performance of the classifier with the remaining 20 % unseen data. To avoid a misclassification of seismic crack events, we defined a high true-positive rate of 0.9
by setting the prediction cutoff threshold accordingly. In our case, all events were classified as crack events if the probability according to the prediction model was 17.2 % or higher. This, of course, lead to a moderately higher number of false-positives

(15 %), but the overall model accuracy was 91 %. The receiver operating characteristic curve (ROC) for the first-step Random Forest model is presented in Fig. S7. The best performance was reached using all available features. Using only selected features did not improve the classifier (see the variable importance in Fig. S8).

Finally, we predicted the class of random 10 % (8,765) of the unclassified events using the classifier trained in the first step. We then manually corrected 1,428 false positives leading to the second larger training data set with 2,072 crack events and 8,037 others. Using this larger data set, we trained a refined Random Forest model following the same steps as described above leading to a smaller false-positive rate of 7 % (see the ROC curve in Fig. S9). The accuracy of the classifier is 94 % which is comparable to previous studies and in the range of human classifiers (Provost et al., 2017; Hibert et al., 2017; Wenner et al., 2021; Langet and Silverberg, 2023). With this refined model we classified all events leading to a dataset of 21,801 seismic crack events and 67,119 others.

## 3.2 Snowmelt modelling

We modelled the amount and timing of snowmelt by simulating the dynamic evolution of the snow cover using the one-dimensional open-source software SNOWPACK (Lehning et al., 1999). With meteorological measurements as inputs, SNOW-PACK is capable of replicating snow microstructure, layering, and its interactions with the surrounding environment. The Bavarian Avalanche Warning Service provided input data, recorded at 10-minute intervals. This includes incoming and outgoing shortwave radiation, snow depth, relative humidity, air temperature, total precipitation, snow surface temperature, and wind speed/direction. Simulations were conducted individually for Zugspitze and Nebelhorn and for each hydrological year from October 1 to September 31, using 15-minute time steps. The measured snow depth functioned as a proxy for precipitation inputs, influencing the mass balance. To filter erroneous measurements, the internal MeteoIO pre-processing library was employed. At both stations, we included a constant ground temperature of 0°C and at Nebelhorn an albedo estimation from shortwave radiation, as these values are not measured at the stations.

The model parameters (see Section S2) were adjusted for each simulation to best fit the melting phase, although discrepancies between modelled and measured snow heights remained possible due to model limitations. The main output for this study was the quantity of snowmelt, expressed in $\mathrm{kg\,m^{-2}}$, which represents the amount of liquid water flowing from the snow cover into the ground. As the Hochvogel is situated between Nebelhorn and Zugsitze (in position and altitude), we used the mean snowmelt of the two sites for further analysis, except for the melting season 2021, where Nebelhorn data were not available.

## 3.3 Synoptical time series analysis

To correlate deformation (10 min), rainfall (10 min), snowmelt (1 h), seismic (100 Hz) and temperature (10 min) data, we aggregated all data to hourly values (original data frequency in brackets) and created common time series plots (Fig. 4 and 3). For better comparability we used derived rates for all variables except the temperature data set (e.g., deformation rate, rainfall intensity and snowmelt in $\mathrm{mm\,h^{-1}}$, crack rate in $\mathrm{events\,h^{-1}}$ and temperature in °C). Due to the strong instrumental and diurnal noise in combination with partly very low signal rates, we smoothed all curves to uncover the measured process dynamics. We used smoothing window lengths with centered running means between 1.5 and 7 d depending on the SNR and the observed

process (see captions of plots for individual values). The use of centered windows implicate the inclusion of future data at any given time, but it smooths the measured data without creating a lag and is applied uniformly to driver and effect. Columns in each sub-plot represent 12 h non-smoothed means.

Based on an apparent correlation of variable pairs, we have identified three types of proxy behaviour: deformation during rain events, displacements in the snowmelt phase and thermally driven rock cracking. Within the complete time series, we identified 15 focus time periods highlighting the relationship of the particular variables. For each focus period we performed a detailed analysis (Fig. 5 to 10 and Fig. S10 to S18), including a time series plot of the two selected variables to illustrate their correlation (subplot a). Mathematically, we tested the correlation via a cross-correlation analysis of the two curves (subplot b). Here, we only considered positive time lags between the driver (e.g., rain) and the result (e.g., deformation). For the time lag with the highest correlation coefficient, we then created a scatterplot with accordingly shifted data and fitted a linear least squares regression (subplot c).

To extend the analysis over the complete available time series, we additionally performed a running cross-correlation analysis. For two selected variables, we calculated the cross-correlation function on a subset of the data with window lengths between 20 and 60 d depending on the observed process. We iterated this by moving the analysis window in 1 d steps and plotted the correlation coefficient for several time lags against the time (Fig. S19 to S25).

## 3.4 Newmark displacement and topographic amplification

We calculated the Newmark displacement (a theoretical index displacement of slopes during seismic acceleration) using a widely used regression model with the formula for slope failures after Jibson (2007, Eq. 9):

$$log\ D_N = 2.401\ log\ I_a - 3.481\ log\ a_c - 3.230 \pm 0.656 \tag{1}$$

where $D_N$ is the Newmark displacement in cm, $I_a$ is the Arias Intensity in $\mathrm{m\,s^{-1}}$ and $a_c$ is the critical acceleration in terms of $g$ (the gravity constant). The critical acceleration $a_c$ in $\mathrm{m\,s^{-2}}$ depends on the factor of safety (FOS, ratio of resisting forces over driving forces) of the area of interest and the sliding plane's slope angle $\alpha$ in degrees (Newmark, 1965; Jibson, 1993):

$$a_c = (FOS - 1)\ sin\ \alpha \tag{2}$$

The Arias Intensity can be estimated after the formula (Wilson and Keefer, 1985; Jibson, 1993):

$$log\ I_a = M - 2\ log\ \sqrt[2]{D^2 + h^2} - 4.1 \tag{3}$$

where M is the earthquake magnitude, D is the epicenter distance in km and h is the focal depth in km.

The exact FOS at the Hochvogel instability can currently not be determined due to uncertainties in the location and condition of the sliding surface. Due to the ongoing mass movement over many decades (Leinauer et al., 2020), we assume only few remaining rock bridges in the carbonate mass and infer a FOS close to failure about 1.1 (Knapp et al., 2018; Heckmann et al., 2012). To incorporate uncertainty, we further evaluated a variety of FOS values where a FOS of 1.05 represents conditions closer to failure, 1.2 less sensitive conditions and 1.01 a slope instability that is imminently failing. We calculated all theoretical

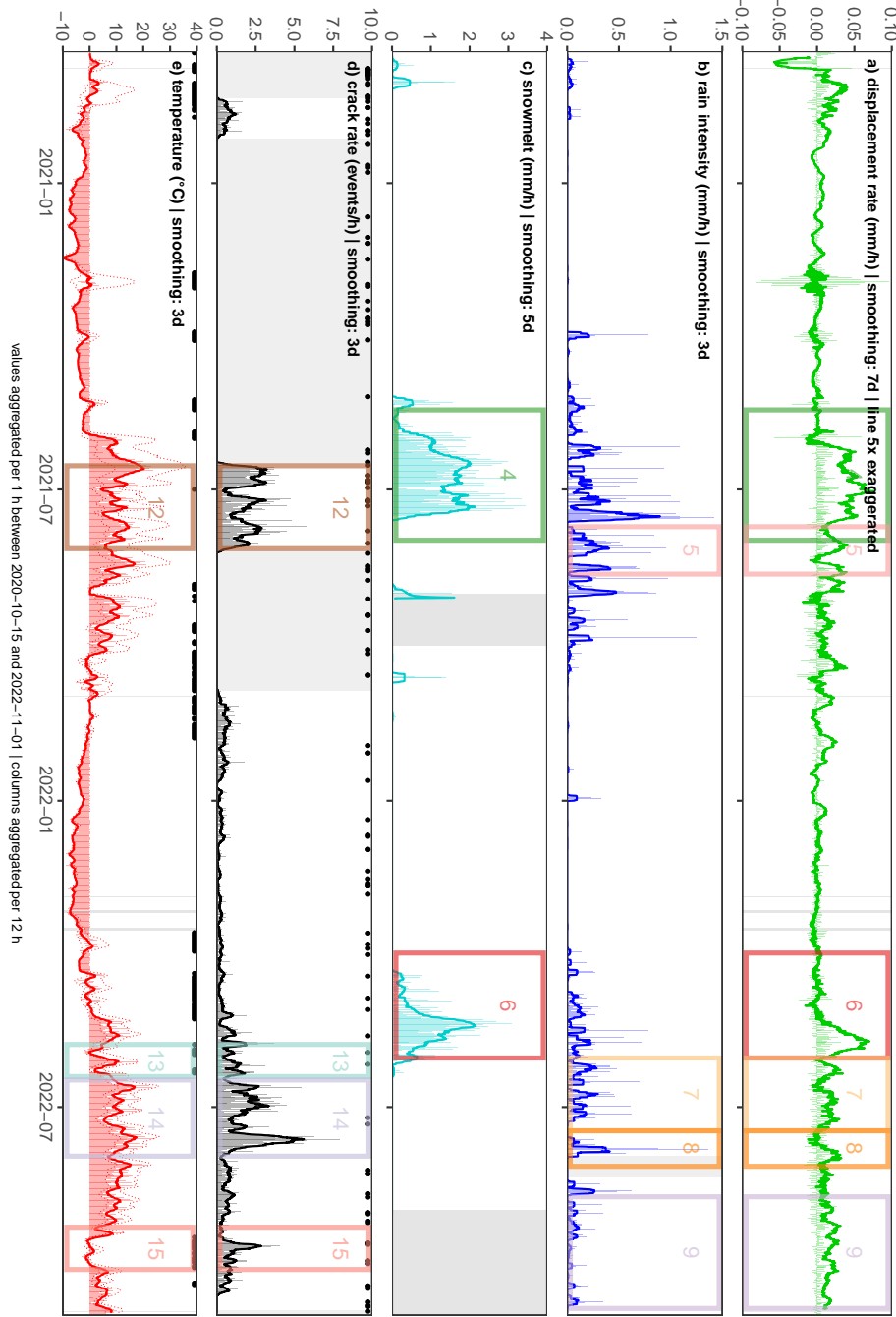

**Figure 3.** Analysed data between Oct 2020 and Nov 2022 with marked and numbered focus times (rectangles). Data are aggregated to 1 h resolution (see the degree of smoothing in the headers). Columns give 12 h means. (a) displacement rate (mm/h), (b) rain intensity (mm/h), (c) snowmelt (mm/h), (d) seismic crack rate (events/h), black dots mark the timing of earthquakes from the catalogue, (e) mean temperature (°C), dashed lines give min and max values, black dots mark days with freeze-thaw/ thaw-freeze conditions.

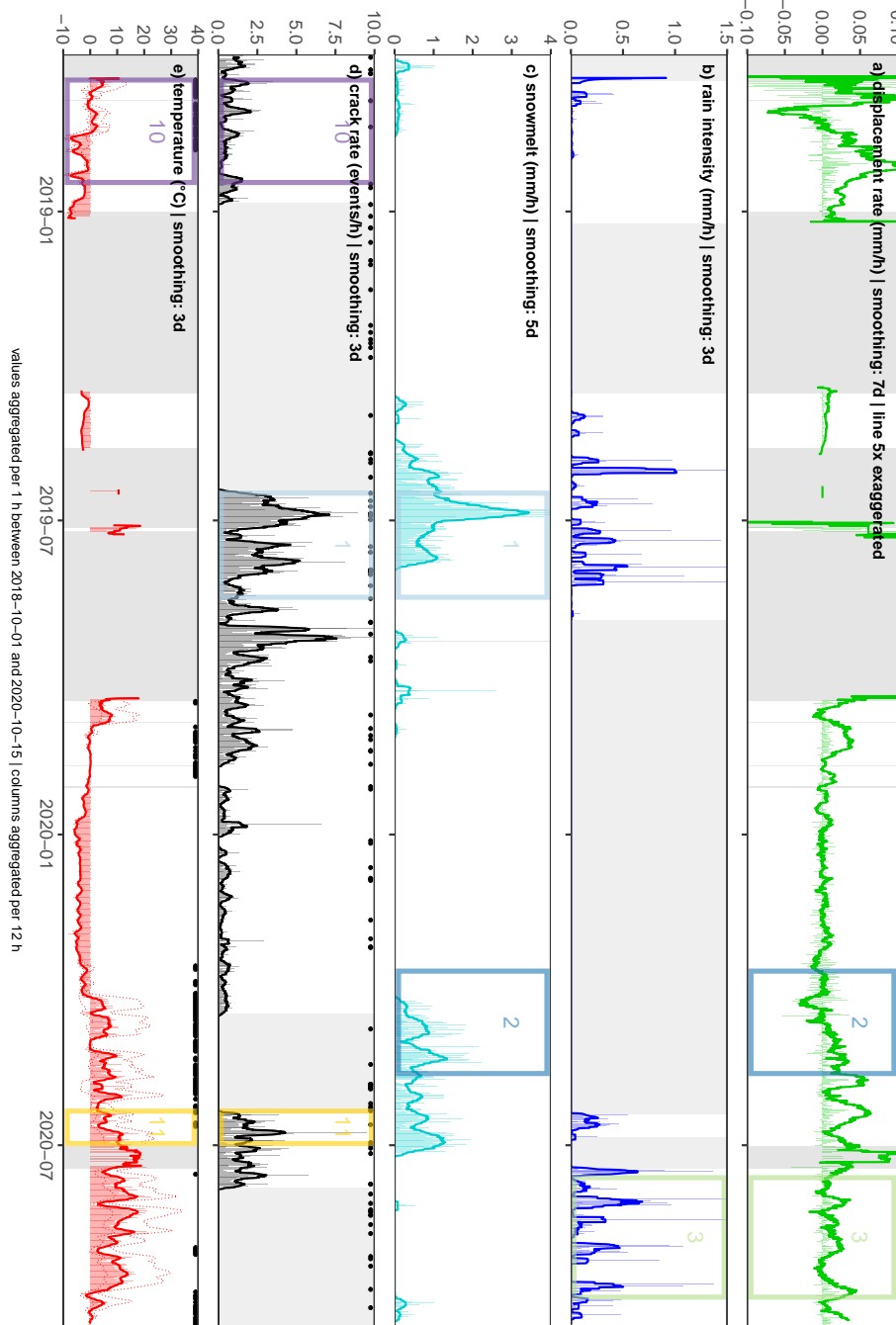

**Figure 4.** Analysed data between Oct 2018 and Oct 2020 with marked and numbered focus times (rectangles). Data are aggregated to 1 h resolution (see the degree of smoothing in the headers). Columns give 12 h means. (a) displacement rate (mm/h), (b) rain intensity (mm/h), (c) snowmelt (mm/h), (d) seismic crack rate (events/h), black dots mark the timing of earthquakes from the catalogue, (e) mean temperature (°C), dashed lines give min and max values, black dots mark days with freeze-thaw/ thaw-freeze conditions.

Newmark displacements for the set of FOS between 1.01 and 1.2, slope angles between 25° and 85°, magnitudes up to $M_w = 8$, distances up to D = 150 km and a focal depth of h = 8 km (mean depth in the earthquake catalogue). We assume a conservative critical Newmark displacement of 2 cm above which rockslides can be triggered (Miles and Keefer, 2001; Meyenfeld, 2009). In a magnitude-distance plot, this critical displacement appears as line per FOS (for a fixed slope angle, Fig. 11 and Fig. S28 to S34). To asses the possible effect of typical earthquakes in the Hochvogel region, we filtered the BGR and SED catalogues for all events with epicenter distances of less than 150 km from the Hochvogel and plotted these into the magnitude-distance plot. More distant events are considered to have no major effect. The ten events with the biggest Newmark displacements have been assessed further including the uncertainty contained in Formula (1) using their exact focal depth (Fig. S35 to S41).

To assess the effect of topographic amplification, we use the peak ground velocity (PGV) measured by our seismometers after several earthquake events (see example record in Fig. S42). It is often complex to distinguish between topographic resonance effects and interacting localized site effects (Rault et al., 2020; Weber et al., 2022) but landslide appearance and rockfall volume correlate with high peak ground accelerations (Meunier et al., 2007; Massey et al., 2022). During earthquakes, the stability of slopes relies on the magnitude of ground motion and its frequency content (e.g. Jibson et al., 2000; Rault et al., 2020). We therefore look at PGV values as the phenomenologial result of seismic stimulation measured by our sensors. We compare the measured PGV during earthquakes at the summit station $HV_1$ with the stations $HV_2$, $HV_3$ and $HV_5$ at the flank lower in the valley and the station $SA_{23}$ on the stable side of the summit. The station $HV_4$ had to be excluded due to the insufficient number of recorded earthquakes. We used all earthquake events from the BGR and SED catalogues that have been recorded on all three components (Z, N, E) of each particular station (Fig. S27). For comparison of regional and distant earthquakes, we additionally used 18 events with a distance D > 15,000 km and magnitude $M_w > 6$ from the USGS catalogue. We then detected the PGV in the signal envelope in 1 Hz windows moved in 0.25 Hz steps between 0.5 and 10 Hz for all components, stations and earthquake events. The ratio of the PGV at station $HV_1$ against the other stations (site-to-reference ratio) is used as indicator of amplifying effects.

## 4 Results and discussion

### 4.1 Rainfall induced displacement

During the warm summer months after snowmelt (June or July to October) the rock mass shows accelerated movement in connection with strong precipitation events. During this period, 38 % of the total crack opening happens during wet days although these only account for 26 % of the total time. The average displacement rate is 1.8 times higher during wet periods compared to periods without precipitation. Looking at peak velocities, this effect reaches a factor of 4-5. Determining the significance of the overall rainfall effect is complicated by the superposition of various effects over parts of the time series: (i) rainfall induced displacements, (ii) simultaneous snowmelt and rainfall, (iii) rainfalls without obvious acceleration, (iv) accelerations without obvious reason, and (v) non-measured precipitation due to the unheated rain gauge. We thus focused on the periods where the displacements are clearly induced by rainfall only.

The direct comparison of rainfall intensity and displacement rates during the selected focus times reveals a high correlation with a time lag of 1-16 h (Fig. 5 and 6, and Fig. S10 and S12). In the respective subplot (b), the blue dashed lines give the values beyond which the correlations are significantly different from zero, which is clearly the case for all focus time periods. Without superposition with other effects like snowmelt, this behaviour can be observed in more than 20 rainfall events across the four summers. The lag likely accounts for the rainfall infiltration time and until maximum hydrostatic pressure is built in the discontinuities within the rock mass. Both depend on pre-event saturation as we cannot observe a consistent change in lag time throughout each season. In addition, the saturation of the discontinuity with water might eliminate the joint cohesion in some discontinuities, lower the joint friction angle especially in the basal marl layers and reduce the effective normal stress at the sliding surface due to buoyancy (Erismann and Abele, 2001), leading to increased displacement rates.

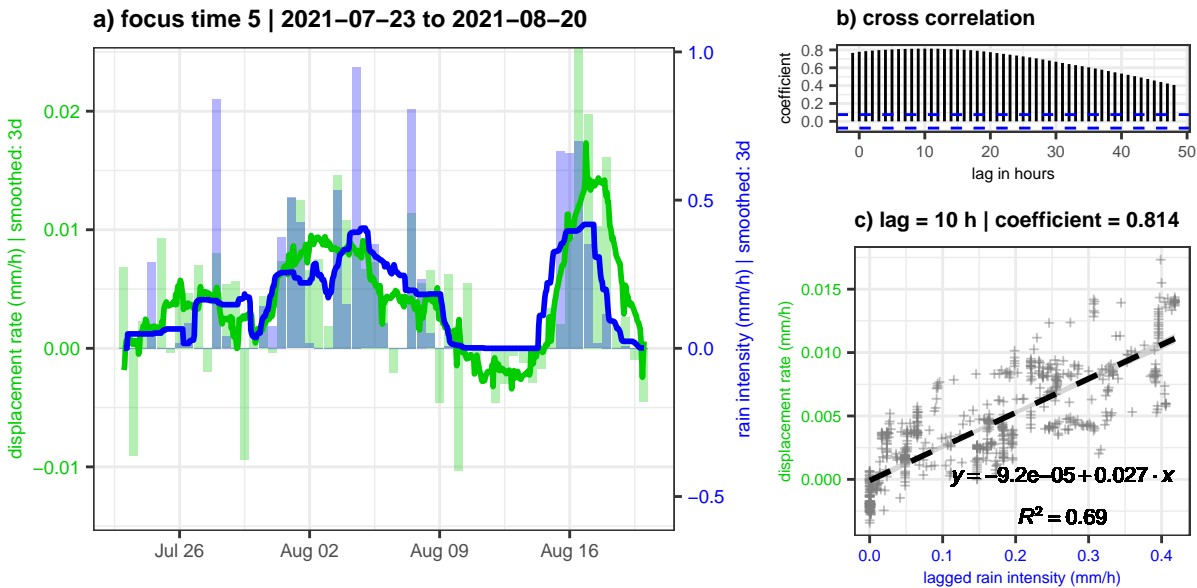

**Figure 5.** Detail plot of focus time 5. (a) displacement rate and rain intensity (lines 3 d smoothed, columns 12 h means). (b) cross-correlation coefficient of the two lines. The highest correlation appears with a lag of 10 h and a coefficient of 0.814. (c) scatter plot with linear trendline (95 % confidence interval as grey area) with 10 h shifted data.

We cannot identify an activation threshold, meaning that even small rainfalls can accelerate the mass movement (see focus time 9, Fig. S12). Within dry periods, timely well-constrained intense rainfall events with several $\mathrm{mm\,h^{-1}}$ accelerate the mass up to 1 $\mathrm{mm\,d^{-1}}$ (cp. August 16 in Fig. 5). When the water input decreases at the end of the rain event the unstable mass decelerates. There is no permanent sealing of discontinuities and perched water drains within several hours. On the other hand, rain events that happen close together within few days result in a common velocity peak due to the retention of water in the system (cp. focus time 8 in Fig. S11). This proposes an immediate lowering of water saturation at least in the fractures in case of dry conditions but also that a proportion of the infiltrated water stays in the system for several days. This can precondition the system in case of further water infiltration, as partially to fully saturated conditions promote the build-up of hydrostatic

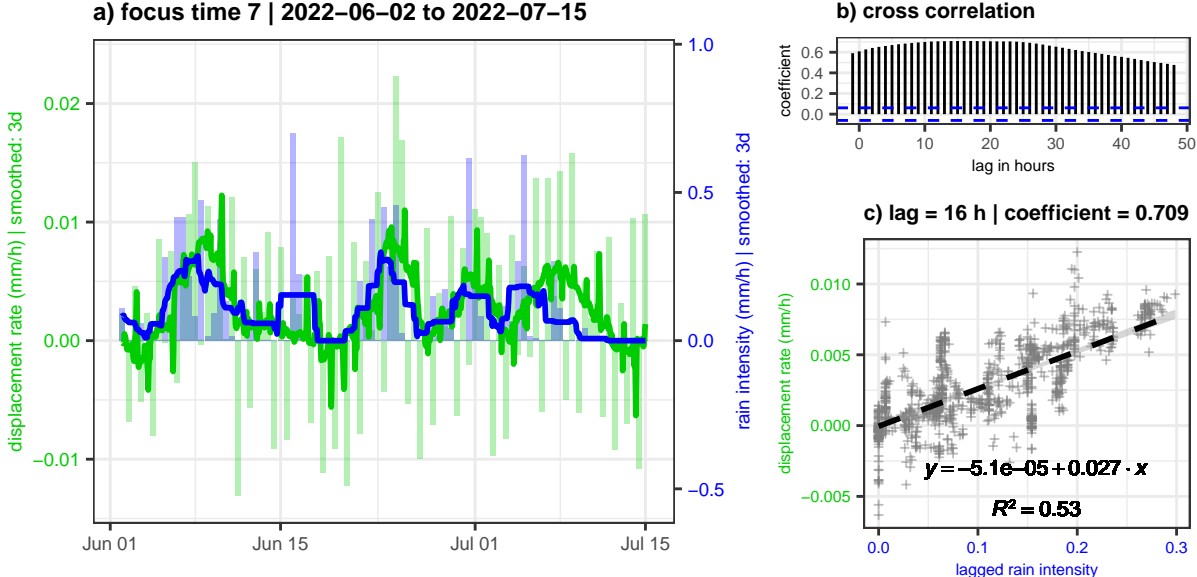

**Figure 6.** Detail plot of focus time 7. (a) displacement rate and rain intensity (lines 3 d smoothed, columns 12 h means). (b) cross-correlation coefficient of the two lines. The highest correlation appears with a lag of 16 h and a coefficient of 0.709. (c) scatter plot with linear trendline (95 % confidence interval as grey area) with 16 h shifted data.

pressure. However, an inert reaction of the unstable mass in the form of low velocity changes necessitates relatively long smoothing windows, which might in turn outsmooth short-duration rain events. Then, the 12 h columns help to interpret the process and a good balance between smoothing window length, event duration and SNR must be applied.

From the linear regression between rain and displacement we infer that in general rain intensities of 0.3-0.6 $\mathrm{mm\,h^{-1}}$ trigger displacement rates of ca. 0.01 $\mathrm{mm\,h^{-1}}$ at the Hochvogel. The running cross-correlation (Supplementary Fig. 20) gives high correlation coefficients of >0.75 with small lag times each summer after snowmelt. This supports the interpretation of a rainfall controlled regime during the snow-free summers. However, the correlation coefficient fluctuates due to the short duration of rain events. The generally strong and immediate response of the rock slope to rain events is indicative of existing substantial damage and a high criticality of the slope (Gischig et al., 2016).

### 4.2 Snowmelt induced displacement

Snowmelt usually occurs between April and July and contributes significant amounts of water to the system, causing accelerated slope movements (Fig. 7 and 8, and Fig. S14). While melt water generally affects slope dynamics the same way as precipitation, our cross-correlation analysis suggests longer lag times, between 4 and 9 days. This is likely related to the slower but therefore more continuous supply of water into the rock mass. Moreover, as there is no snow station directly at Hochvogel, the snowmelt amount is modelled based on measurement data from neighbouring peaks, Nebelhorn and Zugspitze. Given that

the south-oriented slopes of the outstanding Hochvogel peak become snow-free quite early, while significant amounts of snow remain in the >10 m deep main fracture much longer, we anticipate differences in the snowmelt characteristics.

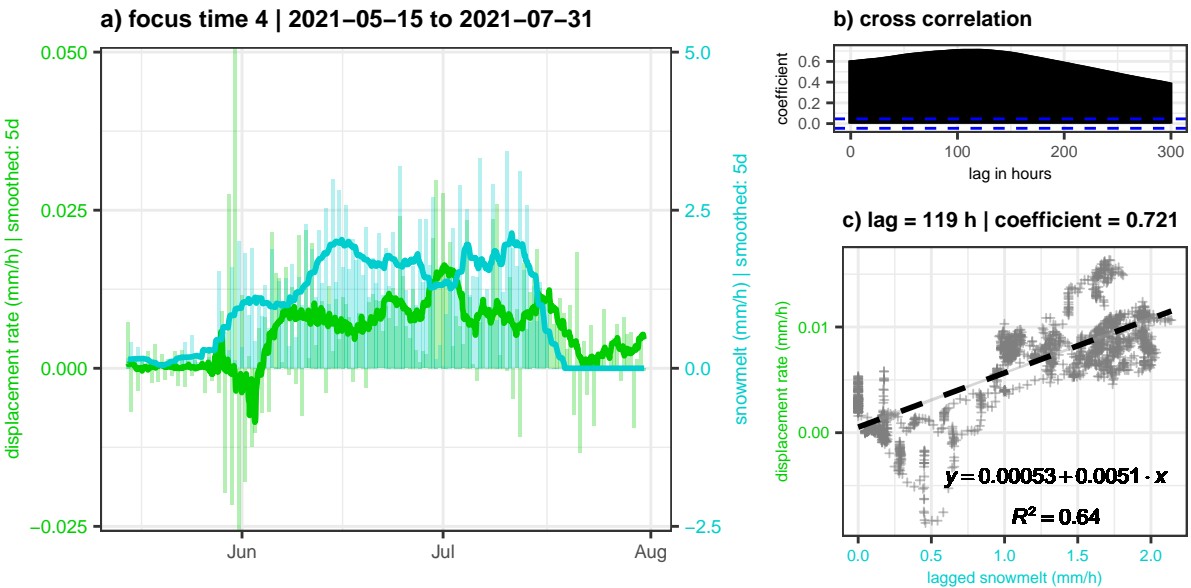

**Figure 7.** Detail plot of focus time 4. (a) displacement rate and snowmelt (lines 5 d smoothed, columns 12 h means). (b) cross-correlation coefficient of the two lines. The highest correlation appears with a lag of 5 d and a coefficient of 0.721. (c) scatter plot with linear trendline (95 % confidence interval as grey area) with 119 h shifted data.

The intensity of the modelled snowmelt generally surpasses the measured rain from the summit, potentially not reflecting the accurate volume of snowmelt infiltrating into the discontinuities at Hochvogel. However, our analysis indicates that, on average, modelled snowmelt of about $1.5 \mathrm{~mm\,h^{-1}}$ corresponds to displacement rates of ca. $0.01 \mathrm{~mm\,h^{-1}}$. Secondary peaks with higher

snowmelt rates inducing temporary accelerations are visible in the velocity curve too (e.g., July 2-15 in Fig. 7 and May 20 in Fig. 8). During early summer, intense rainfall from the first thunderstorm cells superimpose with the late snowmelting phase, making it challenging to distinguish the driving processes effectively during this period. However, rain falling on snow can result in higher water infiltration than from the rainfall alone (Stock et al., 2013).

The running cross-correlation (Suppelementary Fig. 21) gives correlation coefficients above 0.75 with lag times of 4-9 d

each year during snowmelt. This supports our interpretation of a meltwater-controlled regime during the snowmelt season. The snowmelt correlation coefficient fluctuates less than with the rain data, as the snowmelt appears more continuous than the distinct short rain events.

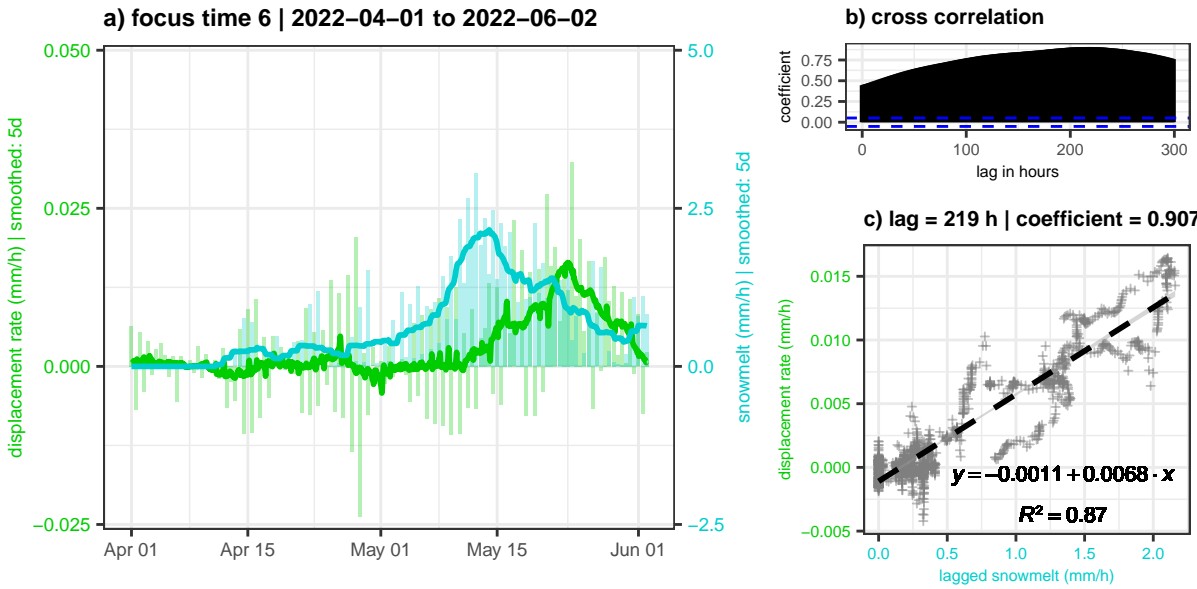

**Figure 8.** Detail plot of focus time 6. (a) displacement rate and snowmelt (lines 5 d smoothed, columns 12 h means). (b) cross-correlation coefficient of the two lines. The highest correlation appears with a lag of 9 d and a coefficient of 0.907. (c) scatter plot with linear trendline (95 % confidence interval as grey area) with 219 h shifted data.

## 4.3   Seismic crack events

The summit network recorded 21,801 discrete events that fall into the class of rock cracking. Their duration was $1.4^{+0.5}_{-0.4}$ s
(median and quartile range), preferentially occurring during daytime. On annual scale, the crack rate is higher during the summer months coinciding with higher displacement rates, higher rain fall intensities and higher temperatures (Fig. S6). We therefore analysed if rock cracking rates and displacement rates interact directly, or if the correlation is rather indirect meaning that environmental forcing increases both, crack rates and slope movement.

We could not find specific time periods where the smoothed crack rate correlates well with the smoothed displacement rate
(Fig. 3 and 4). Likewise, the running cross-correlation analysis (Fig. S19) did not reveal stable high correlation coefficients with a specific time lag. This implicates that there is no obvious correlation between crack rate and displacement rate.

The same applies for the correlation between crack rate and rain fall intensity (Fig. S23). In the melting period 2019 (focus time 1, Fig. S13), we found a high correlation between 7 d-smoothed snowmelt and crack rate at a time lag of 40 h. However, this relation did not emerge in other years and it remains unclear if strong snowmelt can induce enhanced rock cracking.
In contrast, we found the crack rate peaking during temperature peaks (c.p. Fig. 9 and Fig. S15 and S16), likely related to thermal forcing through volumetric expansion and contraction of the rock mass and its minerals. Even small oscillations are represented in both curves. The maximum correlation coefficient appears with a time lag of 0-15 h. Additionally, we observed peaks of rock cracking activity during days with freeze-thaw and/or thaw-freeze conditions (Fig. 10 and Fig. S17 and S18).

Furthermore, the first deep frost of the season without thawing conditions seems to enhance the crack event rate for 4-5 d

(e.g., November 16 and December 11 in 2018, Fig. S17). Here, stress is caused by cryogenic processes in ice-filled fractures or pores, volumetric expansion/ contraction and/or ice segregation (Weber et al., 2017).

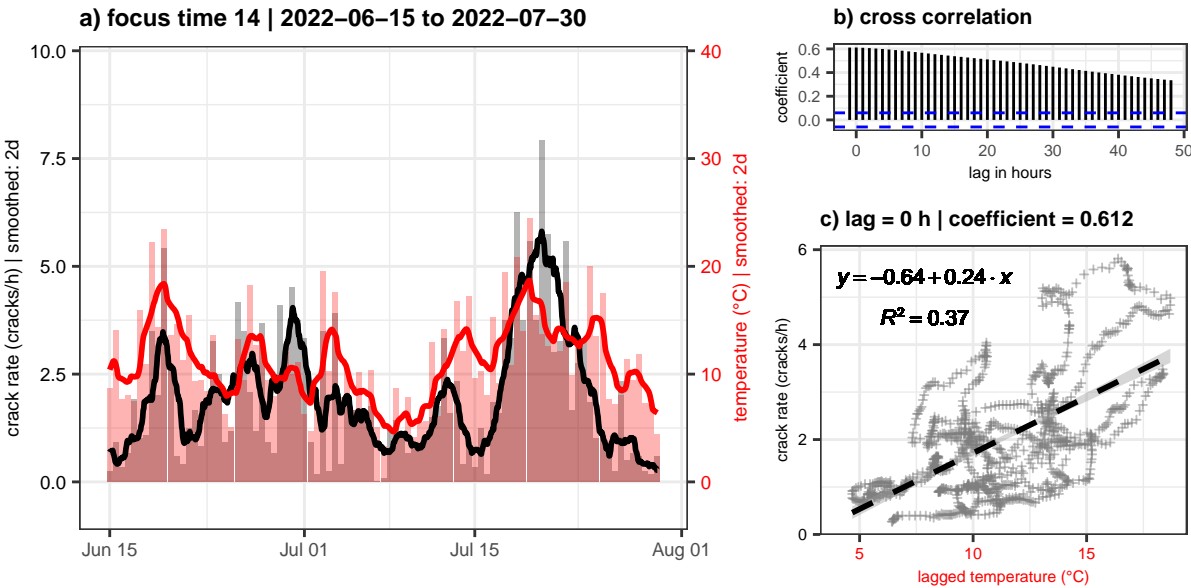

**Figure 9.** Detail plot of focus time 14. (a) crack rate and mean temperature (lines 2 d smoothed, columns 12 h means). (b) cross-correlation coefficient of the two lines. The highest correlation appears without any lag and a coefficient of 0.612. (c) scatter plot with linear trendline (95 % confidence interval as grey area) with data not shifted (0 h).

Heat conduction in rock is relatively slow in the range of $1\ \mathrm{cm\,h^{-1}}$ (Weber et al., 2017; Mulas et al., 2020). The fast reaction of rock cracking activity towards air temperature changes in turn means that most of the crack events that we detected happened close to the surface. This is in line with the results of Dietze et al. (2021). The thermally induced stress can only

affect surface-near rock mass (Bakun-Mazor et al., 2013) unless advective heat transport by percolating water or air can act in fractures (Blikra and Christiansen, 2014; Weber et al., 2017). The shallowness of the detected fracturing events can explain why there is no obvious correlation between crack and displacement rates. Displacements can only result from cracking if rock bridges fail at the sliding plane which we expect to be >10 m away from the rock surface. We assume, that we also detected rock fracturing originating from the sliding plane but the frequency of those events is obscured by the dominant surface-near

crack events. Lagarde et al. (2023) state that crack signals might not always be intense enough to distinguish them from other ambient seismic noise. However, this likely changes when the instability develops closer to failure and the rock fracturing frequency and intensity at the active sliding surface increases. More pronounced rock deformation in the final stage of failure will concurrently intensify the shallow cracking activity. As this activity is then less dependent on temperature changes, it might still be possible to detect a precursory change in activity. On the other hand, crack rates could be altered imminently before

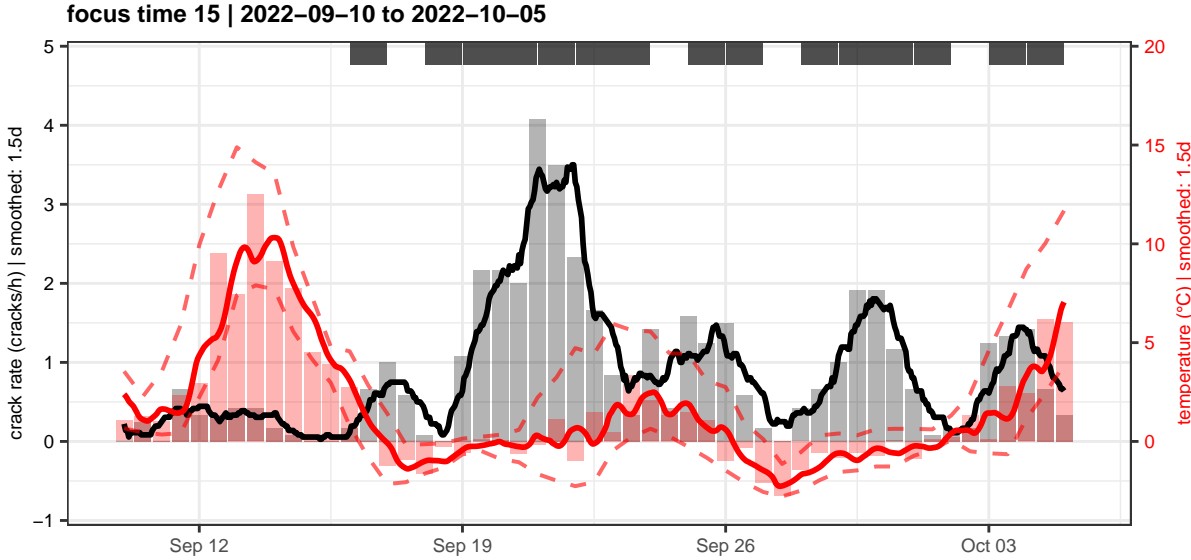

**Figure 10.** Detail plot of focus time 15. Crack rate, mean temperature (solid line), minimum and maximum temperature (dashed lines, all lines 1.5 d smoothed, columns 12 h means). Peaks in the crack rate coincide with days with freeze-thaw or thaw-freeze conditions (black bars on top). The cross-correlation is not shown due to the missing correlation of both curves.

failure, when no rock bridges are left (Lagarde et al., 2023), yet it remains unclear if this can be observed at a multi-block rockslide with the size of the Hochvogel.

### 4.4   The effect of earthquakes and topographic amplification

The analysis of the earthquake catalogue yielded coincidence of a major earthquake with a major rockfall at the Hochvogel in one case only, on June 30, 1935, three days after a M 5.2 earthquake in Bad Saulgau in a distance of 103 km (Hutter, 2010).

Potentially, several historical rockfalls at the Hochvogel might not be documented. An event-specific analysis is therefore not productive and thus we included the whole available earthquake database into our analysis (Fig. S26). It shows that a low-energy background activity is present, mainly originating in the tectonically active alpine valleys, but close high-magnitude events are rare (only 2 events $> M_w$ 5.5 since 1900). This frequent seismic activity could contribute to a low-level promotion of slope instabilities through seismic fatigue. Gischig et al. (2016) could show that earthquake-induced displacements can vary

by more than two orders of magnitude depending on pre-existing damage. However, they used a model with several events with magnitudes $> M_w$ 5.7 and shorter distance. For the Hochvogel, a direct comparison of earthquake timings (black dots in Fig. 4d and 3d) with crack or deformation rates did not show obvious significant influence.

    To evaluate the potential of earthquakes in the Hochvogel region to immediately trigger a major rockfall event, we conducted the Newmark analysis. The calculation of all theoretical Newmark displacements in the magnitude-distance plot (Fig. 11) illus-

trates how strong an earthquake must be at what distance to have the potential to trigger a rockslide. From the >5,000 historical earthquakes in the database, only very few events are in the range of having triggering potential for very low FOS <1.03. A further analysis of the ten events from the catalogue with the biggest Newmark displacements including the uncertainty contained in formula (1) confirms this implication (Fig. S35 to S41). Thus, typical earthquakes around the Hochvogel (e.g. $M_w$ 5 in 20 km distance or $M_w$ 6 in 100 km distance) might only be able to trigger a major rockfall, if the unstable mass is anyways very close to failure (FOS around 1.01). As we interpret the current FOS to be between 1.05 and 1.1, only an exceptionally

strong earthquake (e.g. $M_w$ 6 in 15 km distance or $M_w$ 7 in 50 km distance) could immediately trigger a major failure. Recurrence intervals for such events in this area are estimated to vary between 1,000 and 2,000 years (Oswald et al., 2022). The previously identified 1935 Bad Saulgau event was too weak to have a clear triggering potential. The biggest, but still very moderate displacement is indicated by the historic 1930 Namlos earthquake ($M_w$ 5.3-5.5, 20 km distance). This event induced

enhanced mass wasting processes recorded in local lakes (Oswald et al., 2022), but did not trigger any high-magnitude slope failures (Oswald et al., 2021).

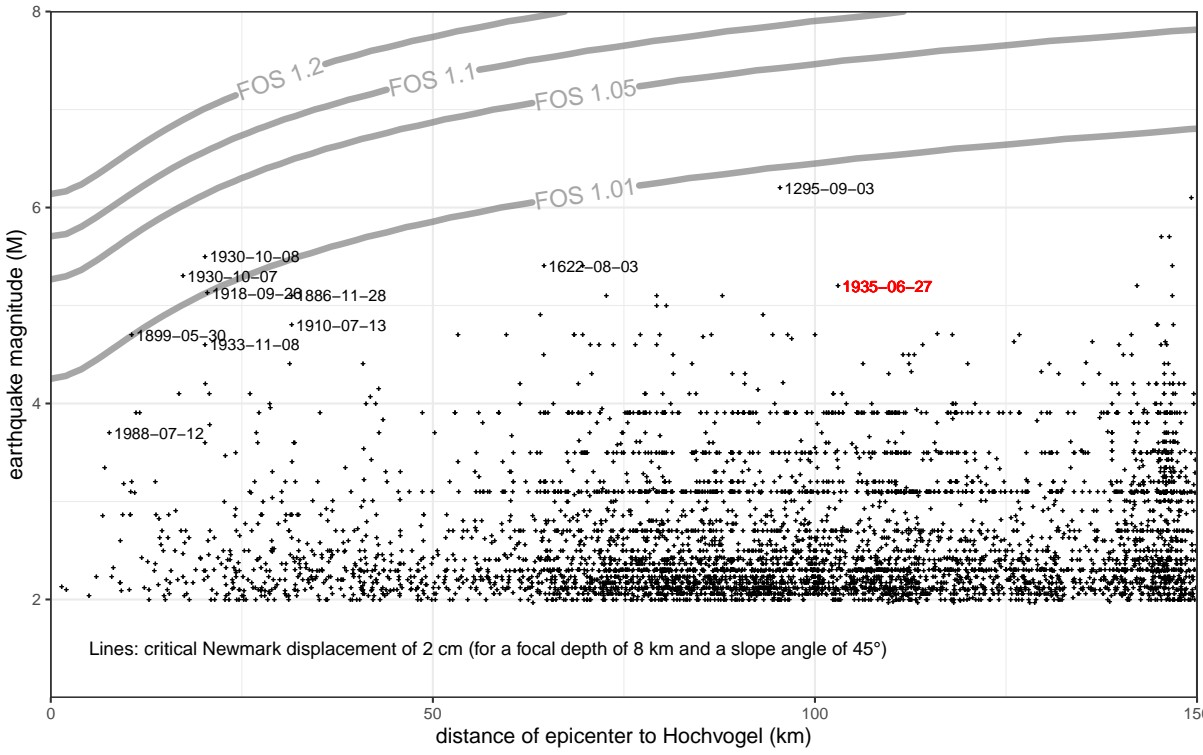

**Figure 11.** Lines indicate for different factors of safety, at which magnitude and distance of an earthquake a theoretical Newmark displacement of 2 cm is expected. This calculation is based on the mean focal depth of 8 km and the mean slope angle of the Hochvogel SW flank (45°). All earthquakes from the catalogues are plotted with black crosses. The earthquakes with the 10 biggest Newmark displacements are labelled in black with their dates. The Bad Saulgau 1935 event is labelled in red. Plots for all other slope angles are in the Supplementary Material.

The evaluation of low earthquake triggering potential holds, although we used a conservative threshold for the critical Newmark displacement of 2 cm. Other studies suggested even higher thresholds between 2-15 cm (Wilson and Keefer, 1985; Jibson et al., 2000; Miles and Keefer, 2001; Meyenfeld, 2009). Additionally, we checked the regression estimate of the Arias intensity with earthquake signals that could be recorded with our local stations. In theses cases, the formula (3) after Wilson and Keefer (1985) overestimates the Arias intensity about 1-2 magnitudes compared to the direct determination after Arias (1970) with the measured ground motion values from our stations. But still, the calculated Newmark displacements are below the threshold.

However, this evaluation could change under the influence of topographic amplification. The geomorphological shape of the Hochvogel massif as well as the location of the unstable mass at the ridge generally favour the effect of topographic amplification (Meunier et al., 2008; Lee et al., 2009a; Khan et al., 2020; Rault et al., 2020). A comparison of the measured peak ground velocity (PGV) during earthquakes at the summit station $HV_1$ against the lower stations shows up to 11 times higher median PGVs at the summit (mean = 3.2, Fig. 12). The extent of this effect varies between the horizontal ($E_{mean}$ = 4.5, $N_{mean}$ = 3.4) and vertical components ($Z_{mean}$ = 1.7), which is typical for topographic amplification (e.g. Bakun-Mazor et al., 2013; Burjánek et al., 2012; Weber et al., 2022), but it remains consistent across the various stations at the valley flank. The horizontal components show the first low-frequency amplification factor peak between 1.5-3 Hz, which is probably around the fundamental resonance frequency of the Hochvogel mountain. Weber et al. (2022) determined a fundamental frequency of 1.8 Hz for the Grosser Mythen, a mountain of comparable shape and similar scale. The amplification effect at the Hochvogel is least pronounced when referencing to the highest valley station $HV_5$ (1933 m) that is situated on a ridge (mean of E-component medians: $HV_2$ = 4.5, $HV_3$ = 4.2, $HV_5$ = 3.8). This is in line with the results of Weber et al. (2022) who measured a mean amplification factor of 9 at the Matterhorn summit and 5 at a ridge below. In general, there is no major difference in median PGV ratios between the local and the far earthquakes. A directional analysis of earthquake epicenters is yet not meaningful due to the limited number of recorded earthquakes.

This could indicate that the summit experiences a significant amplification of factor 2-11 due to topographic site effects and resonant amplification. Then, true displacements due to earthquakes are higher than predicted by the theoretical Newmark displacement analysis and thus the triggering potential is higher than previously assumed. Likewise, the importance of seismic fatigue as a promoting driver is increased by topographic amplification. However, PGV ratios are >1 at station $SA_{23}$ too, which is very close to $HV_1$ but on the stable side of the summit. The general theory would expect a similar amplification at the two summit stations and thus a ratio close to 1. In the low frequencies <4 Hz, there is no amplification peak at station $SA_{23}$, other than at the other stations further down the valley flank. This could be a result of similar site effects at the two summit stations, but still, higher PGVs have been recorded at the $HV_1$ station. This might be due to an amplification and polarization within unstable rock mass itself due to the open main and lateral cracks (Gischig et al., 2016). Burjánek et al. (2010, 2012) measured similar amplification factors with strong variations on the slope scale on two slope instabilities. It is possible that the unstable mass of the rock slope instability close to the wide open and deep main fracture reacts stronger to ground motions than the intact bedrock on the stable side of the summit. As we set up all stations the same way, we assume that all stations have a comparable coupling to the ground. Hence, any differences are supposed to be due to different material properties below the

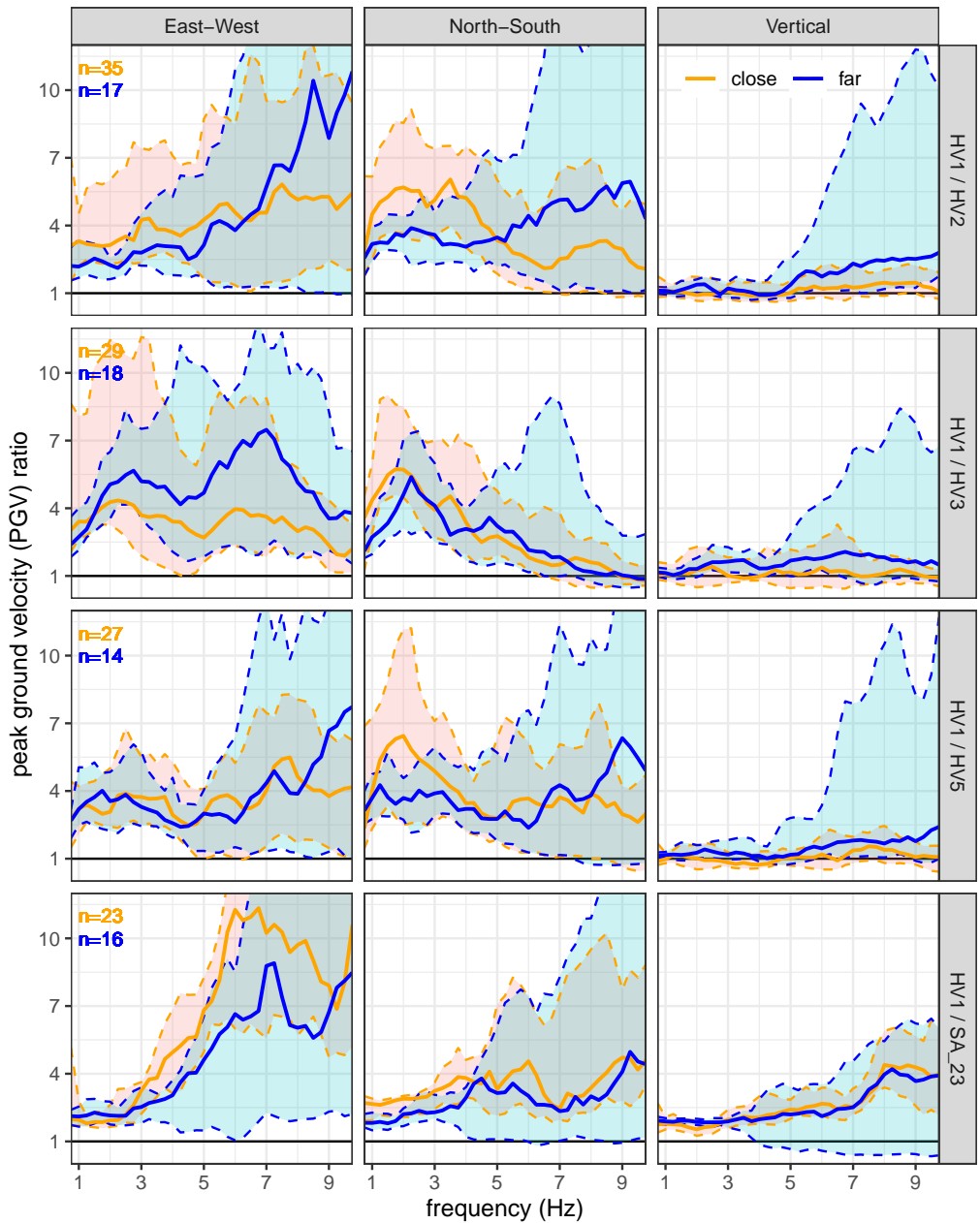

**Figure 12.** Ratio of measured peak ground velocity (PGV) of $HV_1$ on the unstable side at the summit against other stations for close (<150 km, orange) and far (>15,000 km and M>6, blue) earthquakes. The solid line marks the median and the dashed lines the 10th and 90th percentiles. $HV_2$, $HV_3$ and $HV_5$ are located further down the flank, $SA_{23}$ is located on the stable side of the summit.

stations, e.g. a massive rock block at the summit and a weathered layer further down the flank. However, large amplification factors and a strong spatial heterogeneity across the instability due to polarization at fractures may indicate a high criticality of the slope (Gischig et al., 2016). If the seismic amplification and its heterogeneity increases in future monitoring, it could be used as a precursory sign for imminent failure.

## 4.5 Comparative analysis and broader implications

At the Hochvogel, we could analyze the role of multiple drivers and triggers in a comprehensive approach based on continuous real-time data. Previous studies have done this either based on historic event catalogues (e.g. Stock et al., 2013), or smaller-scale rockfalls (e.g. Dietze et al., 2017b). Others investigated aspects of specific drivers at different case sites, that can be compared to the Hochvogel.

The importance of rain water as fundamental driver and trigger is unquestioned (e.g. Guzzetti, 2021; Scandroglio et al., 2021). For example, at Veslemannen in Norway, Kristensen et al. (2021) detected accelerations of a 54,000 $m^3$ slope instability in metamorphic rock due to heavy precipitation events. There, peak velocities appeared 2-24 h after precipitation peaks, similar to the time lags at the Hochvogel. However, at Veslemannen during each seasonal cycle, the reaction of the unstable mass to water infiltration was strongest in autumn, probably due to long-lasting frozen ground conditions that hindered the water to reach the sliding plane. In contrast, the main acceleration phases of instabilities in warmer regions of South and Middle Europe mainly appear in spring or summer due to less deep seasonal frost. The exact timing of the acceleration phase seems to depend on the amount of snowmelt and the timing of the rainy season, but the general pattern is similar among various sites like the Hochvogel, the $8 \times 10^6$ $m^3$ La Saxe rockslide (Crosta et al., 2014) or the $20 \times 10^6$ $m^3$ Ruinon rockslide (Crosta and Agliardi, 2003) in Italy, and the $3.5 \times 10^5$ $m^3$ Preonzo rockslide in Switzerland (Loew et al., 2017). Another study with high temporal resolution has been carried out at the Séchilienne rockslide in the French Alps, where Helmstetter and Garambois (2010) also found no minimum rain threshold for rockfall triggering, and time lags of up to 1 h for rockfalls and a few days for the acceleration of the deep-seated landslide. However, there, accelerated movements lasted for about a month, which might be connected to the bigger size and deeper-seated sliding zone compared to the Hochvogel (50-100 million $m^3$ vs. 200-400 thousand $m^3$).

The driving force of snowmelt has been considered especially in snow-rich countries like Japan (Kawagoe et al., 2009) or Norway (Kristensen et al., 2021). For example, Norway has set up a forecasting and warning service for rainfall- and snowmelt-induced landslides, although this seems to refer mainly to debris and soil mass movements (Krøgli et al., 2018). At Veslemannen, smaller snowmelt events in autumn triggered stronger accelerations than rapid snowmelt in spring (Kristensen et al., 2021). This underlines that a detailed analysis of the exact timing and processual response of rock slope instabilities to snowmelt must be considered. The analysis of the Hochvogel data thus adds valuable insights into timing and slope response to snowmelt for alpine sites with seasonal snow cover.

The analysis of the driving role of temperature changes at the Hochvogel revealed no direct impact on displacement rates, but a strong control of surface-near cracking. This is due to the limited penetration depth of temperature changes on a daily or seasonal scale relative to the depth of the sliding plane. Similar surface-near effects concerning irreversible displacements

or rock fracturing have also been reported by other studies (Bakun-Mazor et al., 2013; Weber et al., 2017; Mulas et al., 2020; Dietze et al., 2021). The only effective way to affect deeper-seated sliding planes seems to be advective heat transport by percolating water or air in fractures (Blikra and Christiansen, 2014; Weber et al., 2017). However, thermal control of displacement rates is likely most important in permafrost slopes (e.g. Mamot et al., 2021).

The local seismicity has a low triggering potential at the Hochvogel rock slope instability. However, other regions receive stronger seismic loading with frequent earthquake-triggered landslides (e.g. Meunier et al., 2007; Massey et al., 2022) and the seismic control varies among different rock slopes. Thus, the effect must always be evaluated site-specifically. This can be achieved with the presented methodological approach of critical Newmark displacement lines per FOS in a magnitude-distance plot (Fig. 11), as these lines are independent of site-specific parameters. Only the local variation of the slope angle

and typical earthquakes must be considered together with topographic site effects. The local amplification factors and patterns at the Hochvogel are similar to those reported on other sites in Switzerland (Burjánek et al., 2010, 2012; Weber et al., 2022), potentially supporting a general pattern. However, the study from Rault et al. (2020) found a complex response of a mountain ridge in Taiwan, illustrating that the seismic response may vary significantly among sites.

     This comparative analysis shows where implications for other sites can be inferred. Water from rainfall and snowmelt, its

amount and its timing, are indeed the most relevant driving factor at high-magnitude instabilities if strong seismicity is absent. The quantification of the driving effects can be transferred to other similar sites, where the geological context, geomechanical behaviour, and climatic conditions are comparable. Disparities exist mostly among rockslides of different volumes and in different climatic settings. Thus, this comprehensive and quantitative study from the Hochvogel can improve the understanding of general and specific process dynamics, and also, where future climatic changes may influence these process dynamics.

**4.6   Climate change effects**

Our results show a clear dependency of slope displacements on infiltrating water from rainfall and snowmelt and of rock fracturing on temperature. Both, temperature and the availability of water, are subject to climatic changes. While rising minimum, mean, and maximum temperatures, are quite evident especially in arctic an high-alpine regions (e.g. Huss et al., 2017; IPCC, 2019; Picarelli et al., 2021), changes in rainfall and snowmelt patterns remain more uncertain and complex.

In some regions, there is a shift towards more intense and more frequent rainstorms (Zhang et al., 2013; Prein et al., 2017; IPCC, 2013, 2014; Pendergrass et al., 2019). This could lead to generally wetter conditions, a higher water table for longer periods, and higher surface and subsurface flows, which in turn leads to a reduction of rainfall amounts needed as trigger, lower cohesion and shear strength, higher hydrostatic pressure and increased erosion (Gariano and Guzzetti, 2016). However, projections on the long-term mean conditions might only be of value for promoting effects, as rock slope failures are more

likely triggered by short to medium duration rainstorms (Krautblatter and Moser, 2009; Picarelli et al., 2021). The northern and western European Alps are projected to become generally wetter, while the southern Alps are becoming dryer, especially in winter and spring (Masson and Frei, 2016). Extreme precipitation events are increasing over large parts of the Alps on annual scale and during all seasons (Ménégoz et al., 2020), but the effect varies significantly between regions and seasons. The increase is shown to be distinct e.g. in autumns in the Southern French Alps (Blanchet et al., 2021), in winter in valleys and medium

mountain areas of the Northern French Alps (Blanchet et al., 2021), or in autumn (Ménégoz et al., 2020) and winter (Frei et al., 2006) in the southern Alps, but other regions show a significant decrease of extreme precipitation events in some seasons. The direct link between climate change and increased rock slope activity is therefore only given, where rainstorm intensity and frequency exacerbate. For the region of the Hochvogel (Northern Calcareous Alps), an increased rainstorm activity has been linked to significantly higher debris flow rates (Dietrich and Krautblatter, 2017; Kiefer et al., 2021), leading to the assumption that the Hochvogel is also receiving an increasing amount of extreme precipitation events.

Future snowfall and melt dynamics are likely even more undetermined and heterogeneous. Higher temperatures are responsible for a shorter and thinner seasonal snow cover and a shifted melting season (Uhlmann et al., 2009; Huss et al., 2017). Overall, snowfall is projected to decrease due to more frequent rainfall-favouring conditions, but, however, cold regions and high altitude mountain areas might experience slight snowfall increases (Frei et al., 2018; Le Roux et al., 2023). This is due to more frequent extreme snowfall events, a general increase in winter precipitation and the shift of very cold areas into more snowfall-favouring temperatures. Overall, this implicates for the Hochvogel that the timing of the main snowmelt phase will likely be shifted towards earlier in spring and that the total amount of snowmelt could decrease. However, extreme snow events and higher temperatures during the snowmelt phase can also lead to more frequent meltwater peaks (cp. Kawagoe et al., 2009) that can accelerate the unstable mass.

## 5 Conclusions

We processed more than four years of high-resolution monitoring data from a very active alpine rock slope instability including displacement, rain, snowmelt, temperature and seismic observations to quantify the main drivers and their triggering potential. This methodological procedure can be transferred to similar cases. At the Hochvogel, the acceleration of the unstable mass even due to small environmental impacts proves a highly sensitive close-to-failure status, at least for parts of the summit.

1. During the *snowmelt phase in spring*, displacements are controlled by meltwater infiltration. The cross-correlation analysis indicates a time lag of 4-9 days between snowmelt input and landslide velocity.

2. During the *snow-free summer*, rainfall controls displacement rates with a time lag of 1-16 h, indicating the possibility of fast hydrostatic pressure increase in the system. Even small rainfalls can accelerate the mass and previous water infiltration preconditions the system in case of consecutive rain events for few days. The rock cracking frequency is mainly controlled by high temperature with a time lag of 0-15 h indicating that we mostly detected surface-near crack events.

3. During the *first frost of the season in autumn*, days with freeze-thaw cycles or clear negative temperatures show higher crack rates. The crack rate might secondarily be controlled by snowmelt or rainfall with low correlation, but it seems not to correlate with displacement directly.

4. During *frozen conditions in winter*, the landslide activity is generally low.

According to our Newmark analysis, recent and historic earthquakes are too weak to have an immediate triggering potential for a major failure at the Hochvogel, unless the FOS is already very close to 1. Seismic topographic amplification of the peak ground velocity of factor 2-11 in the low frequencies is likely and the spatial and frequency-wise heterogeneity of amplification at the summit suggests a high damage and criticality of the slope. In summary, we identified water from rain and snowmelt to be the main driving factor. Accelerations of the slope are mostly connected to high water supply rates or additional water supply in pre-saturated conditions. Comparative to other similar sites, the timing and amount of the water supply controls the driving effect together with geological context, geomechanical behaviour, and climatic conditions. Future increases of rock mass displacements following small environmental impacts as well as changes in the seismic response to earthquakes can express a development towards higher criticality. In the light of ongoing climatic changes that can lead to more frequent and intense heavy precipitation events and shifted snowmelt patterns in many places, our findings suggest that the Hochvogel and similar unstable alpine rock slopes may experience a shift of the environmental forcing dependent on the amount and timing of water supply in the future.

*Code and data availability.* All R-Codes for analysis of the data are available in an online repository together with displacement, temperature and rainfall data from the summit, modelled snowmelt and derived seismic crack event statistics under https://doi.org/10.5281/zenodo. 10567098 (Leinauer, 2024). Snow station data can be obtained from the Bavarian Avalanche Warning Service (Lawinenwarnzentrale im Bayerischen Landesamt für Umwelt). Earthquake catalogues can be accessed via https://services.bgr.de/geophysik/gerseis and http://www. seismo.ethz.ch/en/research-and-teaching/products-software/earthquake-catalogues/ and https://earthquake.usgs.gov/earthquakes/search/. Seismic raw data sum up to 1.09 TB and are available from the authors upon request.

*Author contributions.* JL wrote the manuscript, designed the figures, supervised the real-time monitoring and performed the synoptical data analysis. MD designed and performed the seismic monitoring. JL and MD analysed the seismic data. JL and SK did the earthquake analysis. RS did the snowmelt modelling. JL and MJ performed the rainfall and snowmelt event analysis. MK and MD supervised the study. All authors improved the final version of the manuscript.

*Competing interests.* At least one of the (co-)authors is a member of the editorial board of Earth Surface Dynamics. The authors have no other competing interests to declare.

*Acknowledgements.* This study was developed within the AlpSenseRely project which is funded by the Bavarian State Ministry of the Environment and Consumer Protection (TUS01UFS-76976). We thank all numerous colleagues and friends that helped with field work and technical support during the last 5 years and the Bavarian Avalanche Warning Service for providing original snow station data.

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
