# Peer review of "How water, temperature and seismicity control the preconditioning of massive rock slope failure (Hochvogel)"

_EGUsphere, 2024_

## Referee Comment (RC1)

Review of - *How water, temperature and seismicity control the preparation of massive rock slope failure (Hochvogel, DE/AT)*

The authors present a multi-dataset investigation of rock slope failures at Hochvogel on the German/Austrian border, in particular drawing on displacement, meteorological, and seismic data. The analysis shows an interesting variation in the controls on the slope throughout the seasonal cycle. Overall, I find the manuscript to be well written and the data to be useful for this community and recommend publication after minor revisions.

The only moderate point I would like to raise is around the broader implications of the results – which are currently underdeveloped in this manuscript. This is one impact of the current manuscript structure with combined Results and Discussions – but relatively little substantiative discussion around most points. In particular, it would be good to expand some discussion around how the controls on this slope compare to observations from other regions, and how generalizable your findings might be to other slopes. The case study is interesting but has limited impact alone. Figure 1 begins to touch on some of these elements, but you need to revisit this after presenting your data. I would be open to you keeping the current structure, but perhaps with a new sub-heading discussing these issues, or switching to a separate 'Results' and 'Discussion'. You should bring this into the abstract as well.

I will provide some additional line by line comments:

Title: I am not sure 'preparation' is the correct term. Perhaps 'preconditioning' would be better?

I guess the 'DE/AT' refers to countries, but this is not clear. Either spell the country names out in full or remove it.

L1 – Do we have evidence that the hazard is increasing? I don't see this in the intro. Same for 'exacerbated by climate change' – is this always the case and what evidence do we have? May be worth rewording this first line or adding some info to intro.

L2-3 'comprehensive driver quantifications' is very convoluted and I am not sure what it exactly means here. Is there a reason why 'assessments of preconditioning/triggering mechanisms' or similar doesn't work?

L7 'drives the system' – change to more specific wording.

L8 delete 'Detected'

L12-13 My impression is that this primarily presents and discussed a dataset/monitoring network rather than a methodological approach. If the latter is the intention then some changes are needed.

L13-14 'indicates where climate change […]' Currently, it explores the current controls but doesn't really discuss this in any detail. This is where an expanded discussions section could come in.

L14 'preparation - > 'preconditioning'

L16 From my reading not sure if these refs really show increasing risk from rock slope failures due to climate change.

L19-23 I am missing some reference to the scale mismatch between monitoring data and areas exposed to rockfalls. We are currently very far from having the capacity to instrument all/most hazardous slopes and so rely on other methods. This manuscript can contribute better to this with more discussion of broader implications.

L27 'different time scales' – but then you mention different processes rather than timescales.

L29-30 Presumably a trigger acts at failure, by definition. This sentence is currently a little confusing, consider rewording.

L36-41 This whole paragraph seems unnecessary given the figure. The mention of climate change effects is welcome but needs to be developed further / as mentioned could be represented in the figure.

L44 remove 'significant'

L48 'respectively' -> 'specifically'

L61 – why 'prehistorically'?

L81 remove 'massive' and the brackets.

L83 what is 'it' in this sentence. I though it was 'Hochvogel' but a mountain does not have a magnitude.

L96 'Dominant and outstanding' – review the definitions of these, that description doesn't make sense.

L129 I haven't gone through any code, but I appreciate it being online in an easily accessible and well formatted way.

L145 what do you mean specifically by 'jaggedness'?

L170 and around. It would be good for you to present an ROC curve, or at least in the supplement with some more summary statistics in the main text.

L188 What data gaps. How was this used to fill these? Please explain more.

L189 'Where necessary' – how was this determined?

L196 – 'aggregated to hourly' it would be good to note the original data frequencies here.

L200-203 Some discussion of what this does to the lag time calculations,etc later on would be useful. Also, if a centred rather than trailing window future data is being included in any given time.

L209 Fitted a lin regression how – least squares?

L216 Well introduced -> widely used?

L255 could you calculate the statistical significance of this rainfall effect?

L284 Here discusss more broader context, or in a separate section dedicated to this.

L288 'as described above' -> 'as precipitation'

Fig 5-8c Do you p-values account for the fact that your measurements are not independent due to the smoothing procedure?

Fig 9 Again here – these 'tails' in the scatterplot are characteristic of highly autocorrelated data.

L400 There has still been fairly little comparison with other sites and discussion of broader importance here.

Conclusions – this effectively sums up the findings, nicely done.

L424 'climatic changes that lead to more frequent and intense heavy precipitation events and faster snowmelt' You don't show data anywhere that supports this. Climate change can of course also lead to less frequent rainfall. Snowmelt will not be faster if it is too warm to snow. Etc etc – needs to be supported by data/citations if claiming this.

L425 'stronger environmental forcing' – can you be more specific?

Overall, this paper is a welcome addition to the literature.

-Max Van Wyk de Vries

---

## Author Comment (AC1)

*Dear reviewers,*

*thank you all for taking the time to review our article. Please find all responses to each comment below, marked in blue italics.*

*Following your comments, the most important changes to the manuscript include the addition of two new discussion sections "Comparative analysis and broader implications" and "Climate change effects" that discuss our findings further and set them into the general framework. We further adjusted the abstract, introduction and conclusions accordingly.*

**Reviewer Comment #1**

https://doi.org/10.5194/egusphere-2024-231-RC1

*Dear Maximillian Van Wyk de Vries,*

*thank you very much for your time, constructive and detailed feedback, and your valuable comments that helped to further improve the initial manuscript. Please find our replies and comments below.*

Review of - How water, temperature and seismicity control the preparation of massive rock slope failure (Hochvogel, DE/AT)

The authors present a multi-dataset investigation of rock slope failures at Hochvogel on the German/Austrian border, in particular drawing on displacement, meteorological, and seismic data. The analysis shows an interesting variation in the controls on the slope throughout the seasonal cycle. Overall, I find the manuscript to be well written and the data to be useful for this community and recommend publication after minor revisions.

The only moderate point I would like to raise is around the broader implications of the results – which are currently underdeveloped in this manuscript. This is one impact of the current manuscript structure with combined Results and Discussions – but relatively little substantiative discussion around most points. In particular, it would be good to expand some discussion around how the controls on this slope compare to observations from other regions, and how generalizable your findings might be to other slopes. The case study is interesting but has limited impact alone. Figure 1 begins to touch on some of these elements, but you need to revisit this after presenting your data. I would be open to you keeping the current structure, but perhaps with a new sub-heading discussing these issues, or switching to a separate 'Results' and 'Discussion'. You should bring this into the abstract as well.

*Thank you, this is a good point. We added two new discussion sections after the results named "Comparative analysis and broader implications" and "Climate change effects" that discuss our findings further and set them into the general framework. We further adjusted the abstract, introduction and conclusions accordingly.*

I will provide some additional line by line comments:

Title: I am not sure 'preparation' is the correct term. Perhaps 'preconditioning' would be better?

*OK, changed to 'preconditioning' to make it clearer.*

I guess the 'DE/AT' refers to countries, but this is not clear. Either spell the country names out in full or remove it.

*Ok, removed as suggested.*

L1 – Do we have evidence that the hazard is increasing? I don't see this in the intro. Same for 'exacerbated by climate change' – is this always the case and what evidence do we have? May be worth rewording this first line or adding some info to intro.

*Thanks for this comment. We agree and have rewritten the first two sentences of the abstract to: 'The anticipation of massive rock slope failures is a key mitigation strategy in a changing climate and environment requiring a precise understanding of pre-failure process dynamics.'*

L2-3 'comprehensive driver quantifications' is very convoluted and I am not sure what it exactly means here. Is there a reason why 'assessments of preconditioning/triggering mechanisms' or similar doesn't work?

*Yes, we have considered this in the revised first sentence.*

L7 'drives the system' – change to more specific wording.

*Changed to 'rainfall induces accelerations'.*

L8 delete 'Detected'

*Done.*

L12-13 My impression is that this primarily presents and discussed a dataset/monitoring network rather than a methodological approach. If the latter is the intention then some changes are needed.

*This is right. Changed 'methodological approach' to 'in-depth monitoring data analysis'.*

L13-14 'indicates where climate change […]' Currently, it explores the current controls but doesn't really discuss this in any detail. This is where an expanded discussions section could come in.

*This is now discussed in the new section "Climate change effects".*

L14 'preparation - > 'preconditioning'

*Changed as suggested.*

L16 From my reading not sure if these refs really show increasing risk from rock slope failures due to climate change.

*To make this clearer, we changed the first sentence to: 'Massive rock slope failures are an important geomorphic hazard (e.g. Evans et al., 2006; Lacasse and Nadim, 2009) and in the wake of climate change, landslide risk is expected to increase in many regions (e.g. Gariano and Guzzetti, 2016; Picarelli et al., 2021).'*

L19-23 I am missing some reference to the scale mismatch between monitoring data and areas exposed to rockfalls. We are currently very far from having the capacity to instrument all/most hazardous slopes and so rely on other methods. This manuscript can contribute better to this with more discussion of broader implications.

*This is a good thought. We added this aspect in the description of the research gap and rewrote the last part of this paragraph accordingly.*

L27 'different time scales' – but then you mention different processes rather than timescales.

*We replaced 'repeated' with 'multi-year', so now the described processes are in within the time scales 'seasonal', 'multi-year' and 'long-term'.*

L29-30 Presumably a trigger acts at failure, by definition. This sentence is currently a little confusing, consider rewording.

*We replaced 'Imminently before failure' with 'Finally at failure' to make this clearer.*

L36-41 This whole paragraph seems unnecessary given the figure. The mention of climate change effects is welcome but needs to be developed further / as mentioned could be represented in the figure.

*We added more information on the climate change effects and link to the dedicated new discussion section. However, we would like to keep the sentence listing the potential drivers and triggers in the text to have a comprehensive conclusive text which is supported by the figure.*

L44 remove 'significant'

*Done.*

L48 'respectively' -> 'specifically'

*Done.*

L61 – why 'prehistorically'?

*We moved the word 'prehistorical' next to 'large rockslides' as it belongs to the slides from the cited studies, not to the triggering role of the earthquakes.*

L81 remove 'massive' and the brackets.

*We now added the reference Evans et al., 2006 to the text and removed the brackets as suggested.*

*'Massive rock slope failures' is a fixed term, which has been introduced by Evans et al., 2006. It describes "large or unusually large massive rock slope failure[s] but also with reference to the resultant geomorphic and socio-economic impact". We would therefore like to keep this term.*

L83 what is 'it' in this sentence. I though it was 'Hochvogel' but a mountain does not have a magnitude.

*Thanks. It was supposed to refer to 'rock slope instability'. We modified the sentence to make this clear.*

L96 'Dominant and outstanding' – review the definitions of these, that description doesn't make sense.

*Thanks, this was a translation issue. Changed to 'The Hochvogel is an isolated mountain peak with high topographic prominence…'*

L129 I haven't gone through any code, but I appreciate it being online in an easily accessible and well formatted way.

*Thanks!*

L145 what do you mean specifically by 'jaggedness'?

*Changed to 'highly jointed rock mass'.*

L170 and around. It would be good for you to present an ROC curve, or at least in the supplement with some more summary statistics in the main text.

*The ROC curves of first step and the refined model are already presented in the Supplementary Figures S7 and S9. We inserted an explicit reference to the ROC curves in the main text and added the model accuracy in the main text. Other statistics like false positive rate and true positive rate are given in the text.*

L188 What data gaps. How was this used to fill these? Please explain more.

*Thanks for the question. We rechecked the use of the MeteoIO pre-processing library in our case. MeteoIO can interpolate data if a couple of hours are missing, but actually, we didn't use this option here. Data gaps where either too big for this interpolation (May to June 2021) or absent.*

*Instead, the MeteoIO library was only used to filter erroneous measurements (see the configuration file in Supplementary Material S2 under [FILTERS]).*

*We modified the sentence accordingly.*

L189 'Where necessary' – how was this determined?

*This was necessary for both stations (Nebelhorn and Zugspitze), because they have no measurement of ground temperature (see the configuration file in Supplementary Material S2 under [INPUT]).*

*TSG::CREATE = CST*
*TSG::CST::VALUE = 273*

*with TSG being 'temperature surface ground'.*

*Zugspitze has radiation measurements, Nebelhorn not. We therefore generated the incoming short wave radiation with the parametrization ILWR using a clear sky emissivity (it also relies on TA, RH and the local elevation).*

*ISWR::CREATE = CLEARSKY_SW*

*We clarified the sentence accordingly.*

L196 – 'aggregated to hourly' it would be good to note the original data frequencies here.

*The original data frequencies were mentioned in the 'Study site and instrumentation' Section, but we have added them here as well to have all information at sight.*

L200-203 Some discussion of what this does to the lag time calculations, etc later on would be useful. Also, if a centred rather than trailing window future data is being included in any given time.

*We added some discussion on this here and later in the discussion of the results.*

L209 Fitted a lin regression how – least squares?

*Yes, we used a linear least squares regression. Added this information.*

L216 Well introduced -> widely used?

*Thanks, changed as suggested.*

L255 could you calculate the statistical significance of this rainfall effect?

*We added some information on this to the manuscript.*

*Calculating the significance of the overall rainfall effect is complicated by the complex superposition of various effects over parts of the time series: (i) rainfall induced displacements, (ii) simultaneous snowmelt and rainfall, (iii) rainfalls without obvious acceleration, (iv) accelerations without obvious reason, and (v) non-measured precipitation due to the unheated rain gauge. We thus focused on the periods where (i) the displacements are clearly induced (only) by rainfall. These periods are analysed in detail in the focus time periods (l. 205). In the respective subplot (b), the blue dashed lines give the values beyond which the correlations are significantly different from zero, which is clearly the case for all focus time periods.*

L284 Here discusss more broader context, or in a separate section dedicated to this.

*This is now discussed in the new section "Comparative analysis and broader implications".*

L288 'as described above' -> 'as precipitation'

*Thanks. Changed as suggested.*

Fig 5-8c Do you p-values account for the fact that your measurements are not independent due to the smoothing procedure?

*Thank you for this valuable comment. No, they do not. To avoid misinterpretation of the printed p-values, we excluded them from the annotations of the plots and instead, kept the plotting of the 95 % confidence intervals around the linear regression lines and mention this in the figure captions.*

Fig 9 Again here – these 'tails' in the scatterplot are characteristic of highly autocorrelated data.

*This is true, however, the temperature signal and consequently the cracking signal will always have high probability of being autocorrelated due to periodically warmer and cooler phases. While we cannot change this, we still wanted to calculate a relationship between the two variables over a longer time period (in this case 1.5 months), to show the overall effect that higher temperatures tend to correlate with higher cracking rates. Thus, we sticked with the scatterplot routine here. In fact, the slope of the linear regression varies only from 0.14 to 0.24 in the focus time periods 11, 12, and 14.*

L400 There has still been fairly little comparison with other sites and discussion of broader importance here.

*This is now discussed in the new section "Comparative analysis and broader implications".*

Conclusions – this effectively sums up the findings, nicely done.

*Thank you.*

L424 'climatic changes that lead to more frequent and intense heavy precipitation events and faster snowmelt' You don't show data anywhere that supports this. Climate change can of course also lead to less frequent rainfall. Snowmelt will not be faster if it is too warm to snow. Etc etc – needs to be supported by data/citations if claiming this.

*This is now discussed in the new section "Climate change effects" and the sentence is adjusted accordingly.*

L425 'stronger environmental forcing' – can you be more specific?

*Yes, sentence now changed to: '… may experience a shift of the environmental forcing dependent on the amount and timing of water supply in the future.'*

Overall, this paper is a welcome addition to the literature.

*Thank you!*

-Max Van Wyk de Vries

**Reviewer Comment #2**

https://doi.org/10.5194/egusphere-2024-231-RC2

*Dear reviewer,*

*thank you very much for your time, positive feedback, and your valuable comments that helped to further improve the initial manuscript. Please find our replies and comments below.*

This is a really interesting analysis of data from a highly-instrumented rock mass instability, and is undoubtedly an important contribution to understanding the proximal drivers of small-magnitude displacement events in large slope failures. Analyzing multiple drivers in time series with instrumentation on both displacements and cracking is state of the art and the analyses are done carefully using appropriate methods in my view. The conclusions that water supply rate and timing are the key driver of displacements is not exactly surprising but this is shown here in a compelling and quantitative way, in high time resolution. That the cracking detected was mostly near-surface means that its relevance to the displacement problem is limited, but still this is very useful in separating the effects of temperature and water on the overall system. Therefore, most of my comments are meant to clarify the manuscript, which in general is very clearly written and maintains a tight focus on the data and analysis.

*Thank you for this appropriate and positive feedback. We agree that for some experts it might not be surprising that water is the key driver, but proofing this based on data from a real site on a high-resolution level is not trivial. Thus, we agree on the value of the presented data analysis.*

Line 1. a sweeping motivational statement but too general I think, to say that climate change would in any given region always exacerbate instabilities or the hazards from them.

*This is right. This sentence has been revised completely, see next comments..*

Line 2-3. what are "driver quantifications"?

*A comprehensive quantification of promoting and triggering factors. However, this sentence has been revised completely, see next comment.*

The first two sentences are written more densely with jargon and less clearly than the rest of the abstract. Like most abstracts, this one would be stronger if the first two sentences were simply deleted. Get straight to what you did and leave the motivating for the introduction.

*Thank you for this hint. We reworked the first two sentences, and as a compromise, created one sentence in the style of the rest of the abstract. However, we would like to keep one introducing/ motivating sentence to enable better access for readers from outside the specific research field and to highlight the overall importance of our study.*

*The first sentence now is: 'The anticipation of massive rock slope failures is a key mitigation strategy in a changing climate and environment requiring a precise understanding of pre-failure process dynamics.'*

Line 4: is there a less jargon-laden way to say "well-prepared high magnitude rock slope instability"? On first read this geomorphologist wondered what "prepared" meant in context; you're referring to natural preparation of the rock mass for failure, but at this point in the text it could mean well-prepared by people for monitoring or something else.

*Thanks for this comment. We changed this to 'monitoring data from a large rock slope instability close to failure'.*

83-90 "Due to its magnitude, ... four relevant drivers remain..." This seems like it would fit better in the following section that describes the field site

*Thanks, yes, this might also fit to the 'study site' section, however, here we argue, why we exactly look at the four drivers rain, snow, fracturing and earthquakes, and not at all the other potential drivers. We thus think that this information is important to tighten the focus of the manuscript and to deduce the scope of the study, which we express at the end of the introduction. We therefore suggest keeping the (modified) paragraph here.*

86 Wind is a bit too easily dismissed here, given that it can apply stress to an entire mountainside at once, and that subcritical cracking occurs under very low stresses

*Thanks. To our knowledge, the effect of wind is mostly connected to trees with roots (Stock et al., 2013; Dietze et al., 2015) which are absent at the Hochvogel, seismic noise (Lott et al., 2017), and pressure differences at the rock surface (Stock et al., 2013), which likely do not reach a deep-seated sliding zone. We are not aware of a study evaluating the effect of wind on a high-magnitude rock instability.*

*We inserted this argumentation into the manuscript.*

126 US Geological Survey

*Thanks, changed.*

216 Please define Newmark displacement in a sentence (or parenthetical) for those outside your immediate field

*OK, we inserted a short definition.*

305 "matching the conclusions of Dietze et al" delete, this is a discussion point, not results, and is already discussed below in better context

*Done.*

421-422 Perhaps be more specific in this conclusion that it's the input rate and timing of water e.g. intensity/speed of snowmelt - the rate matters, not just whether and how much water is there

*Good point. We added a sentence naming this conclusion specifically: 'Accelerations of the slope are mostly connected to high water supply rates or additional water supply in pre-saturated conditions.'*

424 "In the light of ongoing climatic changes that lead to more frequent and intense heavy precipitation events and faster snowmelt," are these in evidence in your paper, again I think this is too generalized. In the region of Hochvogel, are there specific projections that indicate these changes?

*Thank you. This is now discussed in the new section "Climate change effects" and the sentence is adjusted accordingly.*

**Community Comment #1**

https://doi.org/10.5194/egusphere-2024-231-CC1

*Dear Giacomo Medici,*

*thank you very much for your time and your valuable comments that helped to further improve the initial manuscript. Please find our replies and comments below.*

General comments

Good and well presented research on rock failures with an angle on snowmelt. The manuscript needs some minor/moderate changes before publication. Please, refer to the specific comments to fix the issues.

 Specific comments

Line 1. "Exacerbated by climate change". Please, be more specific at least in the introduction. You need to explain that due to the changes of the climate the rainfall and snow falls distributions may vary over the hydrological years. If this point is valid also the frequency of the avalanches/landslides can change.

*According to the comments of the other reviewers, the first two sentences of the abstract have been modified extensively.*

*Concerning the climate change effects, we added dedicated discussion section on this topic.*

Lines 36-39. "Possible rockfall drivers and triggers... human or animal activity". Please, add relevant literature below that shows how large volumes of water melts from snow in spring in mountainous areas. All areas characterized by rock slope failures.

- Lorenzi, V., Banzato, F., Barberio, M.D., Goeppert, N., Goldscheider, N., Gori, F., Lacchini, A., Manetta, M., Medici, G., Rusi, S. and Petitta, M., 2024. Tracking flowpaths in a complex karst system through tracer test and hydrogeochemical monitoring: Implications for groundwater protection (Gran Sasso, Italy). Heliyon, 10(2).

- Kawagoe, Saeki, So Kazama, and Priyantha Ranjan Sarukkalige. Assessment of snowmelt triggered landslide hazard and risk in Japan. Cold Regions Science and Technology. 58, no. 3 (2009): 120-129.

- Krøgli, I.K., Devoli, G., Colleuille, H., Boje, S., Sund, M. and Engen, I.K., 2018. The Norwegian forecasting and warning service for rainfall-and snowmelt-induced landslides. Natural hazards and earth system sciences. 18(5), pp.1427-1450.

 *Thank you for this useful literature. We have included it in the introduction and the new discussion section 'Comparative analysis and broader implications'.*

Line 94. Please, disclose the 3 to 4 specific objectives of your research by using numbers (e.g., i, ii, and iii).

*Done.*

Lines 96-125. Add detail on the stratigraphy of the study sites. Age of the dolostones (maybe Triassic)? Geological formations or groups? This change represents detail on spatial heterogeneities that you discuss in your manuscript.

*Yes, it is Upper Triasic from the Lechtal nappe. We added this information.*

Lines 96-125. Provide detail on presence of faults that can represent the spatial heterogeneities of the rock that you mention in your manuscript.

*Ok, added as suggested.*

Line 255. The majority of the snowmelt occurs in April in these regions. Thus, "the warm summer months" actually this time incorporates half part of the spring. Please, fix the statement.

*Thanks, you are right (but I think you mean June instead of April). The majority of the snowmelt occurs in May/ June. Thus, we changed summer months to 'June or July until October'.*

Line 399. Please, provide more detail on the geological nature of the spatial heterogeneities.

*The 'spatial heterogeneity' here refers to polarization effects at fractures across the instability. We made this clearer here.*

Lines 443-620. Integrate the three references suggested above on large volumes of water melted from snow in areas affected by rock slope failures.

*We have included them in the introduction and discussion.*

Figures and tables

Figure 2. I would insert the larger view on the left, and smaller one on the right.

*This order of the subfigure was due to the order in the text. We fixed this and changed the position of the two subfigures.*

Figures 3 and 4. There are words on the vertical axes which are un-readable.

*Thanks, we made that bigger and bold.*

Figures 5 to 9. The equation, "R2" and "p" in the third graphs are un-readable.

*We increased the size of the annotation in the plots. The p-value has been omitted, see review #1.*

---

## Author Response (AR2)

Dear editor,

thank you very much for your positive feedback. We have checked the manuscript and especially the changed parts and corrected typos and grammar in the final uploaded version.

Thanks,

Johannes Leinauer on behalf of all co-authors

Original Editor comment:

The original article was reviewed by 2 experts, and received additional open discussion comments on the topic. Overall, these comments concurred that this was an interesting and worthwhile study on the important topic of rock slope failure. A useful dataset and findings have been delivered, and these will contribute well to the literature. The reviewers raised some important points in need of clarification, while calling for a more complete overview of the work in the context of other studies and the implications of the findings.

Having reviewed the referee comments, the authors reply, and based on my own reading of the revised version, I am satisfied that the comments raised during the review process have been addressed by revisions made. The new additions of text are helpful. These will need some checking through for typos and grammar, but otherwise the review process has improved the paper.